# Indane-1,3-Dione: From Synthetic Strategies to Applications

**DOI:** 10.3390/molecules27185976

**Published:** 2022-09-14

**Authors:** Corentin Pigot, Damien Brunel, Frédéric Dumur

**Affiliations:** Aix Marseille Univ, CNRS, ICR, UMR 7273, F-13397 Marseille, France

**Keywords:** indanedione, chemical modification, domino reaction, MCR, spiro compounds

## Abstract

Indane-1,3-dione is a versatile building block used in numerous applications ranging from biosensing, bioactivity, bioimaging to electronics or photopolymerization. In this review, an overview of the different chemical reactions enabling access to this scaffold but also to the most common derivatives of indane-1,3-dione are presented. Parallel to this, the different applications in which indane-1,3-dione-based structures have been used are also presented, evidencing the versatility of this structure.

## 1. Introduction

Indane-1-3-dione is among one of the most privileged scaffolds in chemistry, as the derivatives of this structure can find applications in various research fields ranging from medicinal chemistry, organic electronics, photopolymerization, to optical sensing and non-linear optical (NLO) applications. One of its closest analogues, namely indanone, is commonly associated with the design of biologically active compounds [1,2,3]. The most relevant examples in this field are undoubtedly Donepezil, which is still under use for the treatment of Alzheimer’s disease [4], or Indinavir, which is used for the treatment of AIDs disease [5]. Interest for indanone derivatives is notably motivated by the fact that this structure can be found in numerous natural products (Caraphenol B isolated from Caragna sinica [6], Pterosin B isolated from marine cyanobacterium [7], another derivative extracted from filamentous marine cyanobacterium Lyngbya majuscula) [7], sustaining the interest for these compounds [8,9]. Due to the similarity of structure with indanone, indane-1,3-dione is also of high current interest, and this molecule has also been extensively studied as a synthetic intermediate for the design of many different biologically active molecules [10]. Beyond its common use in the design of biologically active molecules, indane-1,3-dione is also an electron acceptor widely used for the design of dyes for solar cells applications, photoinitiators of polymerization or chromophores for NLO applications [11]. As an interesting feature, indane-1,3-dione possesses an active methylene group, making this electron acceptor an excellent candidate for its association with electron donors by means of Knoevenagel reactions [12]. Ketone groups can also be easily functionalized with malononitrile, enabling to convert it as a stronger electron acceptor. In this review, an overview of the different chemical modifications performed on the indane-1,3-dione core is reported. Following the description of the synthetic access to the indane-1,3-dione derivatives, their uses in the design of biologically active molecules and organic dyes for various applications in organic electronics are reported. Finally, in the last part, an extensive scope of applications is detailed.

## 2. Chemical Modification of the Indane-1,3-Dione Core

### 2.1. Synthesis of Indane-1,3-Dione

Indane-1,3-dione can be synthesized following different synthetic procedures. Furthermore, the most straightforward one consists in the nucleophilic addition of alkyl acetate **2** on dialkyl phthalate **1** under basic conditions, enabling to produce the intermediate 2-(ethoxycarbonyl)-1,3-dioxo-2,3-dihydro-1*H*-inden-2-ide anion **3**. Then, by heating under acidic conditions, this intermediate can be hydrolyzed and decarboxylated in situ, producing indane-1,3-dione **4** in ca. 50% yield for the two steps [13,14]. However, several procedures were also reported to access **4** by oxidation of indane **5**, using various oxidizing systems such as *N*-hydroxyphthalimide (NHPI) and *tert*-butyl nitrite (*t*-BuONO) [15], H_2_O_2_ with a Mn catalyst [16], pyridinium dichromate (PCC) in the presence of Adogen 464 and sodium percarbonate (Na_2_CO_3_·1.5 H_2_O_2_) [17]. Furthermore, in these different cases, reaction yields remained often limited (17 and 18% yields) while requiring expensive reagents so that the first procedure remains undoubtedly the most popular one to obtain indane-1,3-dione **4** in acceptable yield (see Figure 1). Recently, a two-step procedure was developed, starting from 2-ethynylbenzaldehyde **6** [18]. By means of a Cu-catalyzed intramolecular annulation reaction, 3-hydroxy-2,3-dihydro-1*H*-inden-1-one **7** was prepared in 87% yield and a subsequent oxidation of **7** with Jones’ reagent enabled to obtain **4** in 95% yield. As alternative, *o*-iodoxybenzoic acid (IBX) can also be used as an oxidant, providing **4** in similar yield (92%) [19]. Several strategies were also developed to convert phthalic anhydride **8** into **4** using diethyl malonate **9** and montmorillonite KSF clay [20] or ethyl acetoacetate **11** in the presence of acetic anhydride and triethylamine [21]. In the case of substituted indane-1,3-diones, two distinct routes were developed depending on the substituents attached on the phthalic anhydrides. Thus, in the case of electron-withdrawing groups such as nitro and chlorine (**12–14**), the condensation of malonic acid in pyridine proved to be a straightforward route to access **19–21** [22]. Conversely, in the case of electron-donating groups such as alkyl substituents (**22, 23**), a specific procedure was developed, consisting [23] in a Friedel–Craft reaction of 4-methylbenzoyl chloride **22** or 3,4-dimethylbenzoyl chloride **23** with malonyl dichloride **24**, followed by an acidic treatment with concentrated hydrochloric acid furnishing after purification of the two compounds **25** and **26** in 33 and 15% yield, respectively (see Figure 2). 

### 2.2. Chemical Engineering around the Ketone Groups

#### 2.2.1. Functionalization with Cyano Groups

2-(3-Oxo-2,3-dihydro-1*H*-inden-1-ylidene)malononitrile **28** [24,25,26,27,28] and 2,2′-(1*H*-indene-1,3(2*H*)-diylidene)dimalononitrile **29** [28,29,30] can be synthesized by a Knoevenagel reaction of malononitrile **27** on **4** in ethanol using sodium acetate or piperidine as the bases. In the two cases, an excess of malononitrile was used, and the selection between the di- and the tetracyano-substituted derivative could be obtained by controlling the reaction temperature. Thus, the dicyano compound **28** can be obtained at room temperature contrarily to the tetracyano one **29** that is synthesized by heating the reaction media. In the case of **28**, reaction yields ranging between 61 and 85% were determined, whereas **29** could be obtained with reaction yields ranging from 34 to 45% yield. Functionalization of substituted indane-1,3-diones (**30**, **32**, **34**) with only one dicyanomethylene group was also examined, and several situations were found. Thus, the Knoevenagel reaction furnished a mixture of inseparable isomers **31,31′**, **33,33′** and **35,35′** when 5-methyl-1*H*-indene-1,3(2*H*)-dione **30** [23], ethyl 1,3-dioxo-2,3-dihydro-1*H*-indene-5-carboxylate **32** [31] or 5-fluoro-1*H*-indene-1,3(2*H*)-dione **34** [32] were used as the starting materials (see Figure 3). 

Conversely, 5-alkoxy-1*H*-indene-1,3(2*H*)-diones **36a** furnished selectively **37a** in 63% yield as a result of the specific activation of one of the two ketones by mesomeric effects [33]. A similar behavior was also observed during the synthesis of **39** [32], **41** [34] and **43** [35], their precursors **38**, **40** and **42** being substituted with halogens. The steric hindrance generated by the substituents was another strategy to control the regioselectivity, and the synthesis of **45** is a relevant example of this (see Figure 4) [36]. Concerning the symmetrically substituted indane-1,3-diones, those bearing halogens at the 5,6-positions were the most commonly studied, as exemplified with compounds **47** [37] or **49** [32,38].

#### 2.2.2. Self-Condensation of Indane-1,3-Dione: The Bindone Adduct

Bindone **50** is an electron acceptor widely used due to its stronger electron-withdrawing ability compared to that of **29**. It can be easily synthesized by self-condensation of indane-1,3-dione **4** in basic conditions (triethylamine [25], sodium acetate [39], sodium hydride [40]), or in acidic conditions (sulfuric acid [41]) (see Figure 5). 

#### 2.2.3. Formation of *bis*-Thiazoles and *bis*-Thiazolidinone

Indane-1,3-dione **4** and corresponding derivatives have been extensively studied for their biological activities ranging from antitumor, antibacterian and anti-inflammatory activities [13,42,43,44,45]. Parallel to this, 1,3,4-thiadiazole groups also exhibit biological activities [46,47,48,49,50,51,52,53,54] so that their combinations were examined [55]. From a synthetic viewpoint, *bis*-thiazoles could be obtained in two steps starting from indane-1,3-dione **4**. By first reacting **4** with hydrazinecarboxamide **51** in ethanol, in the presence of triethylamine, 2,2′-((1*H*-indene-1,3(2*H*)-diylidene)*bis*(hydrazine-1-carboxamide) **52** could be obtained. Then, upon reaction with a number of *N*-aryl-2-oxopropane-hydrazonoyl chloride derivatives **53–55**, *bis*-thiazoles **56–58** could be isolated with reaction yields ranging from 78 to 89%. Using a similar procedure, reaction of **52** with ethyl (*N*-arylhydrazono)chloroacetate **59–61** could furnish the corresponding *bis*-thiazolidinone **62–64** in high yields (79–90%) (see Figure 6).

### 2.3. Chemical Engineering around the Aromatic Groups

#### 2.3.1. Polyaromatic Structures

To improve the electron-accepting ability of **4**, an alternative to the substitution of indane-1,3-dione with malononitrile consists in developing polyaromatic structures. Notably, naphthalene derivatives were prepared, following the same synthetic route to that used for **4**, consisting in the condensation of ethyl acetate **66** on diethyl naphthalene-2,3-dicarboxylate **65** in solvent-free and basic conditions. After decarboxylation, **67** could be prepared in 91% yield for the two steps [12,56,57,58,59,60]. Introduction of lateral groups onto **67** is possible but involves a specific route to be developed. A Diels–Alder reaction between l,3-diphenylbenzo[*c*]furan **69** and cyclopent-4-ene-l,3-dione **70** furnishes the 1,3-diphenylbenzo[*c*]furan-cyclopent-4 ene-1,3-dione adduct **71**. By dehydration in acidic conditions (HCl/H_2_SO_4_), **71** could be converted to **72** in 25% yield. Recently, the design of a helical-shaped structure **75** was reported [36]. If the structure is innovative, the synthesis is identical to that used for **4**, starting from dimethyl naphthalene-1,2-dicarboxylate **73** (see Figure 7).

#### 2.3.2. Halogenated Indane-1,3-Diones

Halogenation of indane-1,3-dione derivatives subsequent to their synthesis is not possible such that such derivatives can only be obtained by first introducing halogens onto their corresponding precursors. Notably, as a first synthetic approach, halogenated phthalic anhydrides were converted as indane-1,3-diones using ethyl acetoacetate, and a series of halogenated indane-1,3-diones **76–82** is presented in Figure 8 [20,21,28,37,38,40,61]. Parallel to this, the strategy previously mentioned that AlCl_3_-promoted acylation of a benzoyl chloride derivative (**83**) with malonyl chloride **24** proved to be another effective approach to design chlorinated indane-1,3-dione derivatives (**84**) (see Figure 8).

#### 2.3.3. Introduction of Various Electron-Withdrawing Groups on Aromatic Ring

##### Nitration

As previously mentioned for halogenation, electrophilic aromatic substitution cannot be carried out on indane-1,3-dione **4** such that a post-functionalization with nitro groups is required. To date, only few indane-1,3-diones bearing nitro groups have been reported in the literature (see Figure 9) [21,22,62].

#### 2.3.4. Cyanation

To the best of our knowledge, no cyano-substituted indane-1,3-dione derivatives have been reported to date. Furthermore, such acceptors are as crucial as the cyano groups, which are among the best electron-accepting groups.

#### 2.3.5. Introduction of Alkoxy-Carbonyl Groups

Here again, only the post-functionalization of naphthalic anhydrides was used to introduce CO_2_R groups. An example is provided below with **32** (see Figure 10) [31]. By using ethyl acetoacetate **11**, anhydride acetic as the solvent and triethylamine as the base, **32** could be obtained from **32a** in 71% yield.

### 2.4. Chemical Engineering around the Methylene Group

#### 2.4.1. Knoevenagel Reaction

Due to the presence of the two ketones groups on both sides of the methylene groups, indane-1,3-dione **4** possesses a privileged group for realizing Knoevenagel reactions. In the case of indane-1,3-dione **4** and its substituted derivatives, the condensation reaction can be carried out in the conditions initially used by Knoevenagel in 1894 to condense benzaldehyde with ethyl acetoacetate **11**, namely in ethanol with a catalytic amount of piperidine [61]. Typically, Knoevenagel reactions performed with **4** or **68** can be realized with reaction yields higher than 70% (see Table 1 and Figure 11) [12]. As the main interest of this reaction, use of a highly polar solvent favors the precipitation of dyes **95–114** upon cooling, and the reaction can be carried out in green conditions since a non-dangerous solvent can be used. Additionally, the work-up can be limited to a simple filtration, avoiding the use of complicated purification processes [56,57].

When 2-(3-oxo-2,3-dihydro-1*H*-inden-1-ylidene)malononitrile **28** and its analogues are involved in Knoevenagel reactions, another amine should be used, and diisopropylethylamine (DIPEA), which is a none-nucleophilic base, is the most popular one. As reported in several recent works, an unexpected nucleophilic addition of secondary amines onto the cyano groups of the push–pull dyes can occur, giving rise to a cyclization reaction and producing 3-(dialkylamino)-1,2-dihydro-9-oxo-9*H*-indeno [2,1-*c*]pyridine-4-carbonitrile derivatives **115**, according to the mechanism proposed in Figure 12. 

Numerous examples of undesired cyclization reactions have notably been reported with piperidine providing 3-(dialkylamino)-1,2-dihydro-9-oxo-9*H*-indeno [2,1-*c*]pyridine-4-carbonitrile derivatives **115** instead of the expected push–pull dyes [62,63,64]. In 2019, an unprecedented nucleophilic addition of piperidine on **101** was also reported, providing **102** after dehydration. The mechanism supporting the formation of this unexpected structure is depicted in Figure 13, and the crystal structure of this molecule presented in Figure 1 undoubtedly proved its formation.

When 2,2′-(1*H*-indene-1,3(2*H*)-diylidene)dimalononitrile **29** and its derivatives are engaged in Knoevenagel reactions, the high stability of their anions in basic conditions impedes the Knoevenagel reactions to proceed. Therefore, acidic conditions should be used, and acetic anhydride is commonly used in this aim (see Figure 14) [65,66].

Besides, several reports mention the use of the classical piperidine/ethanol conditions to condense **29** onto aromatic aldehydes, despites the strong deactivation of the **29** anion in basic conditions. A few examples of products obtained in these conditions are presented in Figure 15 (**121** [67], **123** [67] or **125** [68]).

To avoid the use of base, several authors replaced ethanol by solvents of higher boiling points such as methylethylketone (see Figure 16) [69]. By refluxing **29** and **126** at elevated temperature, **127** could be obtained in 42% yield.

#### 2.4.2. Oxidation Reaction

Among indane-1,3-dione derivatives, ninhydrin is well known, as it can be advantageously used as a revelator for thin layer chromatography (TLC). Over the years, several strategies have been developed to access to these structures [70]. For instance, oxidation of indane-1-one **128** with selenium oxide (SeO_2_) can furnish ninhydrin **138** [71], but this reaction was not limited to **4**, and the oxidation reaction tolerates various substituents such as OMe, *tert*-Bu, Me, Br, CF_3_ or NO_2_ (see compounds **139–147,**
Figure 17) [72]. Direct oxidation of **4** was also investigated to convert it, as **138** and ninhydrin **138** could be prepared in 48% yield using the dual oxidizing system SeO_2_/H_2_O_2_ [73] in 40% yield for the two steps using iodobenzene diacetate [74] and 94% yield using *N*-bromosuccinimide (NBS) in DMSO [75]. However, all attempts to oxidize **4** as **138** using sodium hypochlorite failed, and phthalic acid was isolated as the unique product of the reaction [76]. The authors also demonstrated **138** to be oxidized as phthalic acid, evidencing that sodium hypochlorite is a too strong oxidant. The oxidation reaction with *N*-bromosuccinimide (NBS) in DMSO was not limited to indane-1,3-dione **4,** and indane-1-one **149** and indane-2-one **128** could also be oxidized using the same procedure, providing **138** in 82 and 85% yield, respectively. Photooxidation of **4** in the presence of tetrabutylammonium hexafluorophosphate and oxygen using Rose Bengal as the photosensitizer could convert **4** as **138** in 75% yield upon irradiation with a UV light [77].

#### 2.4.3. Halogenation

α,α-Dihalogenation reactions of carbonyl compounds have been extensively studied in the literature [78,79,80,81,82,83,84,85,86,87,88], and different procedures were thus developed for the halogenation of indane-1,3-dione **4**. Notably, traditional reagents of halogenation including *N*-chlorosuccinimide (NCS) or NBS in ethanol could provide **150** and **151** in 95% and 92% yields [89]. Several green syntheses were also developed, all based on mechanical ball milling. Using this approach, halogenating agents such as trichloroisocyanuric acid or tribromoisocyanuric acid could furnish **150** and **151** in 98 and 97% yields [90]. Similarly, mechanosynthesis of **150** could be efficiently achieved by employing sodium bromide and oxone (98% yield) [85] or ammonium bromide and oxone (see Figure 18) [91].

The conversion of nucleophilic halogens to electrophilic ones could be realized by reacting Lewis acids such as ZnBr_2_ or AlCl_3_ with lead tetraacetate [92]. High reaction yields were also obtained during the synthesis of **150** while using KBr/KBrO_3_ (86% yield) [93], 1,3-dibromo-5,5-dimethylhydantoin in acetic acid (88% yield) [94]. Similarly, **151** could be obtained while reacting **4** with 1,3-dichloro-5,5-dimethylhydantoin in acetic acid (89% yield) [94]. Finally, selectfluor^®^ (1-chloromethyl-4-fluoro-1,4-diazoniabicyclo [2.2.2]octane *bis*(tetrafluoroborate)) was the most widely studied fluorinated agent for fluorination of **4** in water while using a surfactant (Genapol LRO) (74% yield) [95], sodium dodecyl sulfate in water (93% yield) [96], or acetonitrile as solvent (60% yield) (see Figure 19) [41,97]. Furthermore, numerous drawbacks concerning the electrophilic fluorination with selectfluor^®^ were reported in the literature. Notably, parallel to the formation of the expected F^+^ cation, formation of radical species (F^•^) by single electron transfer has also been proposed, even if the mechanistic studies have not fully elucidated the mechanism. Nevertheless, formation of an intermediate monofluorination state could be demonstrated [98].

In the case of indane-1,3-dione **4**, fluorination was proposed as occurring by means of an attack of the double bond of enol onto selectfluor^®^, followed by a deprotonation with the resulting diazoniabicyclo [2.2.2]octane species. By iterating the reaction a second time, **152** could be obtained (see Figure 20) [97]. Considering that there is still a lack of efficiency for the α,α-dibromination of 1,3-diketones, the photoredox catalysis was envisioned as a possible alternative to conventional chemistry to improve the selectivity during bromination [99,100,101,102]. Light is also a traceless reagent so that light-promoted chemistry perfectly fits with the concepts of green chemistry. For bromination, light-activated radical reactions are also extensively described in the literature, enabling to efficiently generate bromine radicals.

In 2019, an interesting reaction was developed to accelerate the formation of bromide radicals and, in this aim, *N*-bromosuccinimide (NBS) was irradiated in the presence of bromoacetic acid and under visible light [103]. By this unique combination, two concurrent bromination mechanisms could be evidenced, the first one consisting in an electrophilic bromination resulting from the photoassisted formation of bromine, and the second one consisting in a radical bromination process resulting from the formation of bromide radicals promoted by light (see Figure 21).

By the presence of these double sources of brominating agents, the exceptional reaction yield of 99% could be obtained as well as accounts from the coexistence of both the electrophilic and the radical bromination reactions (see Figure 22).

#### 2.4.4. Cyanation

In 1977, an interesting procedure was developed to introduce a cyano group at the methylene position of indane-1,3-dione. Inspired by the synthesis of indane-1,3-dione **4** starting from dialkyl phthalate and ethyl acetate, an analogue procedure was proposed, where ethyl acetate **51** was replaced by acetonitrile (see Figure 23) [104]. Using sodium methanoate as the base, **153** could be obtained in 78% yield.

#### 2.4.5. Nitration

Nitration of the methylene group of indane-1,3-dione **4** can be performed in one step, by using nitric acid as the reagent [105]. Furthermore, **154** could be obtained in 78% yield (see Figure 24).

## 3. Indane-1,3-Diones as Reagents for Various Chemical Transformations

### 3.1. Synthesis of Cyclophanes

Cyclophanes play an important role in supramolecular chemistry [106,107,108,109,110,111,112,113,114,115,116,117,118,119], especially as hosts for host–guest strategies so that a continuous effort has been made to develop new, efficient and simple synthetic routes to access to these structures [120,121,122,123,124,125,126]. Among the main methods reported in the literature, Wurtz coupling [127] carbene insertion [128,129], Ni-catalyzed Grignard coupling [130], polymerization of *p*-xylene [131], pyrolysis of sulfones [132] or acyloin condensation [133] can be cited as popular reactions. Conversely, Suzuki-Miyaura cross-coupling reaction [134,135,136,137] or ring-closing metathesis [138,139,140,141,142,143,144,145,146,147] have been less studied. In 2012, a combination of these two key reactions (Suzuki-Miyaura cross-coupling reaction and ring-closing metathesis) has been conceived as a concise and efficient synthetic route to access to [4,4]-cyclophane derivatives [148]. In the first step, dialkylation of indane-1,3-dione **4** with *meta* or *para*-bromobenzyl bromide furnished **155a** and **155b.** The functionalization of the methylene group of indane-1,3-dione **4** was not an easy task since conventional alkylation conditions such as NaH in THF totally failed. After several attempts, use of freshly prepared KF-celite in dry acetonitrile enabled to produce **155a** and **155b** in moderate yields, 74 and 70%, respectively. These two intermediates were subsequently functionalized with allyl groups, by a Suzuki-Miyaura cross-coupling reaction using an excess of allylboronic acid pinacol ester. Finally, metathesis reactions of **156a** and **156b** using the first generation of Grubbs catalyst and titanium isopropoxide delivered the two macrocycles **157a** and **157b** in 54 and 45% yields. Finally, hydrogenation using a catalytic amount of Pd(C) yielded **158a** and **158b** in 80% for the two cyclophanes (see Figure 25).

### 3.2. Synthesis of Crown Ether Derivatives of Indane-1,3-Dione

Compounds capable of changing the optical properties by complexation with metal cations are at the center of numerous researches, and in this field, crown ethers consisting in a ring containing ether groups capable to bind alkali cations have been extensively studied [149,150,151]. Considering that indane-1,3-dione **4** is a strong electron-acceptor, its combination with an electron donor connected to a crown ether could make the final assembly an interesting structure for ion sensing. Such a push–pull dye was reported in 2010 by Mitewa and coworkers [152]. Thus, **159** could be prepared by a Claisen–Schmidt condensation of 2-acetyl-1,3-indandione **160** with the appropriate aldehyde **161** in acetic acid. The different attempts to use piperidine as the base failed, and the starting materials **160** and **161** were entirely recovered after reaction. Unexpectedly, the Claisen–Schmidt reaction furnished a fully conjugated molecule, resulting from a deacetylation reaction, according to the mechanism depicted in Figure 26. 

To prepare 2-(1-hydroxyethylidene)-1*H*-indene-1,3(2*H*)-dione **162**, several strategies can be developed, as shown in Figure 27. Notably, a selective cyclo-oligomerization based on a self-condensation of acetyl chloride **163** in the presence of aluminum trichloride AlCl_3_ followed by a cross-condensation reaction with benzoyl chloride **164** could provide **162** in 87% yield according to the mechanism depicted in Figure 27 [153]. Parallel to this, **162** can also be synthesized using the Kilgore procedure [154], consisting in an addition–elimination process of a ketone (acetone **166**) on diethyl phthalate **165** under basic conditions (sodium ethoxide) (see Figure 27) [155].

Finally, examination of the complexation of **159** with various divalent cations revealed no drastic changes of its absorption spectrum upon complexation, except with Sr^2+^ and Ba^2+^. This unexpected result was assigned to the low involvement of the nitrogen lone pair of the crown ether in the coordination of the cations, thus weakly affecting the optical properties of the push–pull dye connected to the crown ether. 

### 3.3. Synthesis of Tetracycline Heterocyclic Analogues

Tetracyclines are an extended family of compounds discovered and developed in the mid-1940s for their promising therapeutic properties [156]. Notably, tetracyclines are efficient as antibiotics and anti-malarial drugs, and these structures have also been identified as being beneficial of various pathologies such as cancers or Parkinson’s disease. Due to the increasing resistance of microbes to antimicrobials used from long ago, analogues to tetracyclines are actively researched, and a series of pentacyclines comprising the indane-1,3-dione motif have been synthesized [25]. The synthesis was relatively straightforward since the combination of malononitrile **27**, salicylaldehyde **167** and indane-1,3-dione **4** in a one-pot procedure furnished 6-amino-7-imino-7*H*-indeno [2′,1′:5,6]pyrano [3,4-*c*]chromen-13(13b*H*)-one **168** in 80% yield. Imine group in **168** could be easily cleaved in acidic conditions, yielding **169** (see Figure 28). 

The one-pot synthesis of pentacyclines was not limited to **4** as the starting material, and another derivative **172** was also prepared starting from **28** following two different synthetic routes in order to confirm its chemical structure (see Figure 29). Thus, following a first step consisting in the reaction of **28** with the diazonium salts **170** and **173**, the two products **171** and **174** were subjected to an intramolecular cyclization reaction by reflux in ethanol in the presence of a catalytic amount of piperidine, providing **172** and **175** in 85 and 87% yields, respectively. Finally, hydrolysis of imine **175** under acidic conditions furnished **172** in 90% yield and confirmed the formation of this compound by the similitude of the ^1^H NMR spectra.

On the basis of the two synthetic routes, other tetracycline derivatives comprising three fused cycles were also prepared, as exemplified with the indenopyrane derivatives **177** and **179** or the indeno-[2,1-*c*]-pyridazines **181** and **182** listed below (see Figure 30).

### 3.4. Synthesis of Indane-1,3-Dione Derivatives via Tert-Butylisocyanide Insertion

Isocyanide insertion into palladium–carbon bonds has emerged as an effective strategy to form C-N, C-O or C-C bonds [157,158,159,160,161,162,163], and numerous derivatives such as quinazolinones [164], 6*H*-isoindolo [2,1-*a*]indol-6-ones or indenoindolones [165] have been prepared using this strategy. In this context, the chemoselective insertion of *tert*-butyl isocyanide **183** to form C-C has been examined to form indane-1,3-dione derivatives starting from 1-(2-bromophenyl)-2-phenylethanone **184** [166]. This reaction tolerated various substituents since good reaction yields were obtained with 1-(2-bromophenyl)-2-phenylethanone substituted with various electron rich and electron deficient groups (see Figure 31). After hydrochloric acid hydrolysis, indane-1,3-diones substituted at the 2-position with various aromatic rings could be obtained with reaction yields ranging from 61% to 75% (**185–192**). Starting from 1-(2-bromophenyl)ethan-1-one derivatives, indane-1,3-dione with aliphatic substitutes at the 2-position could also be prepared (**190, 191**). Finally, tolerance of this reaction to the cyano group was also evidenced (**192**). In this last case, **192** was not hydrolyzed as in the other case.

### 3.5. Copper-Catalyzed Sulfonamidation of Benzylic C-H Bonds

The transformation of a C-H bond into a C-N bond is an important reaction in organic chemistry, and the amination of primary benzylic groups has not been excluded from this interest [167]. Especially, sulfonamidation of primary benzylic groups with primary and secondary sulfonamides could give access to molecules of biological interest. However, methodologies for sulfonamidation of primary benzylic groups remain scarce in the literature [168,169], and most of the reactions only afford dramatic low yields (9–30%) [168,170,171,172,173,174,175,176,177,178,179,180,181,182,183,184,185,186,187]. Conscious of the paucity of methodologies available, Powell et al. proposed in 2010 after substantial efforts, optimized conditions for the sulfonamidation of toluene **193** with *N*,4-dimethylbenzenesulfonamide **194** at room temperature [188]. A reaction yield as high as 70% could be obtained for the synthesis of **196** while selecting the copper catalyst Cu(CH_3_CN)_4_.PF_6_, the oxidant *tert*-butyl 3-(trifluoromethyl)benzoperoxoate **195** and the ligand, namely indane-1,3-dione **4** (see Figure 32). Choice of the ligand was determined as being crucial, other ligands such as bathophenanthroline or 1,3-diphenylpropanedione only allowing a conversion of 33% and 52%, respectively. If indane-1,3-dione **4** was capable to enhance the overall yield, its exact role remained unclear, **4** being only capable to act as a monodentate ligand for copper. The substitution pattern of the oxidant was also determined as being of prime importance, and the presence of electron-withdrawing groups such as the trifluoromethyl group on *tert*-butyl 3-(trifluoromethyl)benzoperoxoate **195** could weaken the perester bond, thereby facilitating the formation of the *tert*-butoxy radical [189].

### 3.6. Michael Addition on β-Substituted meso-Tetraphenylporphyrins

*meso*-Tetraarylporphyrins are extensively studied due to their facile synthesis, high molar extinction coefficients, high photoluminescence quantum yields, excellent photochemical stability, but also for their ability to chelate a wide range of metal cations in their inner core [190,191].

An efficient strategy to finely tune their optical properties consists in the modification of the porphyrin core by means of changing the π-conjugation length [192,193,194], the introduction of peripheral groups [195,196,197,198,199,200,201] or breaking the planarity [202,203,204,205]. In this field, insertion of a nitro group at the β-position porphyrin macrocycle proved to be an effective approach to introduce various substituents on the porphyrin core. Notably, 2-nitro-5,10,15,20-tetraphenylporphyrin **197** and **198** could undergo a variety of nucleophilic aromatic substitution, the macrocycle behaving as a Michael acceptor. While using indane-1,3-dione **4** as the nucleophile, the nucleophilic substitution could undergo a variety of metalated and non-metalated porphyrins (**175** and **176**) according to the mechanism depicted in Figure 33 [206].

As a result of the addition of indane-1,3-dione **4** onto porphyrins, *meta*-chlorins that differ from porphyrins by the reduction of one of the pyrrole ring could be obtained [207,208]. *trans*-chlorins are extensively studied due to their red-shifted absorption compared to their porphyrin analogues, enabling to design dyes with an infrared absorption [194,209,210,211,212,213]. The six chlorins could be prepared with reaction yields ranging from 67 to 87% yield. Lastly, the same authors reported an unprecedented ring-fusion of *trans*-chlorins bearing 1,3-indanedione functionalities. Upon addition of Ni cations inside the porphyrin core, *trans*-chlorins could undergo a skeletal rearrangement of the porphyrin macrocycle, and *trans*-chlorins could be converted to fused metalloporphyrins by elimination of one indane-1,3-dione unit (as shown in Figure 34) [214]. By controlling the reaction time, the metalation of *trans*-chlorins (15–20 min) or the nickel insertion, followed by an indane-1,3-dione elimination and a ring-fusion (3–4 h) furnished in turn the fused metalloporphyrins. To clarify the role of the nickel cation in this mechanism, reflux in DMF of *trans*-chlorins **199-Zn**, **199-Cu**, **199-Ni**, **200-Zn**, **200-Cu, 200-Ni, 201-Zn, 201-Cu, 201-Ni** with nickel acetate converted all *trans*-chlorins to the fused metalloporphyrins (**202-Ni**, **203-Ni** and **204-Ni**). Metalloporphyrins containing other divalent cations were also prepared, by first demetalating **202-Ni**, **203-Ni** and **204-Ni** and then remetalating with the appropriate metal acetates (Co, Cu, Zn).

### 3.7. Synthesis of 1,4-Isochromandione

1,4-isochromandione is an important heterocyclic compound, as this molecule is the starting material for the synthesis of numerous biologically active compounds such as parvaquone, which is an antiprotozoal agent marketed as Clexon [215,216], or atovaquone, which is an anti-pneumocystic agent marketed as Mepron [217]. In this context, various strategies have been developed to access this elemental building block. The first report mentioning the synthesis of 1,4-isochromandione **208** was reported in 1966 by Holt et al. using indane-1,3-dione **4** as the starting materials, and **208** could be obtained in 86% yield [218]. Following this pioneering work, several improvements were performed in order to improve the reaction yield. Typically, by diazotation of **4** with tosyl azide [219], 2-diazo-1*H*-indene-1,3(2*H*)-dione **210** can be obtained with reaction yields ranging from 59% [220] in ethanol to 80% [221] in triethylamine, 88% [222] in THF and finally 93% for the best conditions in ethanol [223]. By treating first **210** in basic conditions, **211** could be obtained, and treatment in a second step with sulfuric acid could furnish **208** in 83% for the last step (see Figure 35) [224,225]. 

### 3.8. Synthesis of Benzofurans by Electrooxidation of Hydroquinone Derivatives

Benzofurans are important compounds, as these structures are widely used for the treatment of cardiac arythmias, and amiodarone is a relevant example of benzofurans used for this purpose [226,227]. However, benzofurans are not restricted to these applications, and benzofurans are also reported as having numerous pharmaceutical applications so that the synthesis of these derivatives has been widely studied [228]. More generally, benzofurans can also be used as fluorescent probes [229], antioxidants or brightening agents [230,231]. In 2015, an electrochemical synthesis of **212** and **213** was reported by Ameri et al. consisting in oxidizing in situ hydroquinones **214** and **215** as benzoquinones in a phosphate buffer (pH = 7), thus acting as a Michael acceptor (see Figure 36 and Figure 37) [232]. Depending on the substitution of hydroquinone, one or two 1,4-michael additions could occur, according to the mechanism depicted in Figure 37.

Compounds **212** and **213** could be isolated in high yields, 86 and 84%, respectively. By combining several electrochemical techniques, the presence of an ECECECEC and of an ECEC mechanism was proven for **212** and **213**, respectively. 

### 3.9. Combination of Knoevenagel Condensation and Michael Addition Reactions

In a purpose of synthetizing complex organic structures involving affordable building-blocks, multicomponent reactions (MCRs) have played an important role over the last twenty years. With the emergence of green chemistry purposes, the synthetic strategies developed to access to complex structures have to be shortened and the use of metal catalysts and organic solvents limited. To address these issues, MCRs constitute a powerful strategy but also an expeditious method enabling to rapidly generate a vast library of molecules by systematically changing one of the three reactants involved in this convergent approach [233]. Several examples of multicomponent reactions making use of indane-1,3-dione **4** as one of these substantial building blocks for MCRs have been reported during the last decades. Typically, MRCs consisted in the combination of Knoevenagel condensation followed by a cyclisation reaction resulting from a Michael addition. Considering the similarity of structures of all these cyclized compounds obtained during these different works in the presence of an identical moiety, i.e., indane-1,3-dione **4**, these structures can be combined under the generic name of “indeno-fused structures” (see Figure 38). From a structure viewpoint, interest for these heterocycles was notably motivated by their interesting properties in medicinal chemistry, these molecules possessing anti-bacterial, anticancer or cardiovascular activities [226,228,234,235]. Biological applications of these different indeno-fused structures are discussed in the section devoted to the different applications of indane-1,3-dione derivatives.

As first examples of MRCs are those that were devoted to the synthesis of quinolinone derivatives (see Figure 38, structures **216**, **217**). In this work, Sandaroos and coworkers used iron triflate (Fe(OTf)_3_) as the Lewis acid catalyst, and the reaction could be conducted in solvent free-conditions [234]. Choice of iron triflate as the catalyst was supported by the weak nucleophilic character of the triflate anion, making the metal cation a stronger Lewis acid. The reactions performed at 90 °C for 4 h could provide the products with reaction yields ranging from 80 to 92% after purification. Control experiments performed without Fe(OTf)_3_ also revealed the MRCs not to proceed, highlighting the crucial role of the Lewis acid in the activation process. The authors could reuse the metal catalyst without any loss of catalytic activity, but no precision about the number of cycles examined is given. In an attempt to optimize the catalytic activity, several other metal triflates such as Zn(OTf)_2_, Cu(OTf)_2_ were examined, but iron triflate remained the most effective one. To obtain a deeper insight into the mechanism, the reaction could be successfully decomposed into two steps, either by mixing indane-1-3-dione **4** and the aldehyde (**225**) but also the aldehyde and the amine (**226**) in the first step (see Figure 39). Although both synthetic pathways remain possible, the expected compound **217b** could be obtained in the two cases.

Iron plays without contest an important role in catalysis, not only as a heterogenous catalyst but also for the design magnetic nanocomposites. Iron can also be used not only for its remarkable reactivity but also for the easy recovery of the metal catalyst. Such a strategy has been reported in an MCR synthesis of indenoquinoline-8-one derivatives. The Lewis acid developed in this work, namely sulfonic acid-functionalized cellulose-coated Fe_3_O_4_ (Fe_3_O_4_@cellulose-SO_3_H) nanoparticles, could be easily removed from the reaction media by use of an external magnet (see Figure 38, structures **218**) [235]. The authors optimized this synthesis be rendering it applicable in solvent-free conditions but also in water media, with sulfonic groups covering the metal particles for water compatibility. Using these magnetic Fe_3_O_4_ nanoparticles, the desired products could be obtained within 5 min. at 40 °C. Additionally, after removal of the magnetic particles with a magnet, the final product could be purified by a simple recrystallization in EtOH. The metal catalyst proved to be reusable, but a reduction of the reaction yield was nevertheless noticed. Thus, by repeating the synthesis of **218d**, the reaction yield decreased from 95 to 82% yield after five runs. Nevertheless, contrarily to what was observed for the **216/217** series, an aromatization of the structure was observed, leading to the formation of 4-azafluorenones (see Figure 38, structures **218**). Here again, the role of iron particles was essential, by activating the aromatization reaction (see Figure 40).

Similarly to the strategy applied for the easy recovery of the Fe_3_O_4_@cellulose-SO_3_H particles, Fe_3_O_4_@NCs/Cu(II) particles have also been developed for the synthesis of another family of indeno-fused structures, namely **219a–219o** [236]. This bio-based catalyst showed remarkable efficiencies in EtOH at 60 °C since reaction yields ranging from 79 up to 97% could be obtained within only 5 min. of reaction. Here again, the recovery was easy since only a magnet and washing with EtOH was required to recover the catalyst in pure form. A good recyclability was found since the catalyst could be reused without significant loss of its catalytic activity, even after 4 runs. Thus, the reaction yield decreased from 95 to 79% yield after 4 runs for **219a**.

Even though these metal catalysts were highly efficient, it is always desirable not to use catalysts or to use catalysts that can operate in homogeneous phase. In this field, several examples of ionic liquids (ILs) have been proposed as green and reusable catalysts (see Figure 33, molecules **220** and **221**) [237,238]. Ionic liquids have been proposed as an interesting alternative to the traditional catalysts in a variety of chemical reactions, as these molecules are often less pollutant than metals and can also act both as solvents and catalysts [239,240,241,242]. In the case of these reactions, ILs have thus a dual role of solvent and catalyst. A first example of IL is 1,2-dimethyl-*N*-butanesulfonic acid imidazolium hydrogen sulfate [DMBSI]HSO_4_, which was not used as a solvent [237]. Reaction conditions were optimized in ethylene glycol, and the best temperature for the MCR was determined as being 120 °C. In these conditions, the reaction was relatively fast since it could be finished within 4 min. Feasibility of the reaction in solvent-free conditions was examined, but lower reaction yields and longer reaction times were found compared to the results obtained in ethylene glycol. Recyclability of the catalyst was also examined, and after three successive runs, no significant reduction of the reaction yield was found. A few years later, another interesting example was proposed with 1-hexyl-3-methyl-imidazolium iodide [HMIM]I as the ionic liquid. It may be mentioned that in this case, the reaction could be performed in water with high reaction yields (up to 95%) while using sonication as the activation mode [238]. Sonochemistry is not widely used in organic chemistry to activate a large variety of organic reactions due to its appealing features: shorter reaction times, higher reaction yields, less byproduct formed, milder reaction conditions. Even though the kinetic is a bit slower than with the previous IL [DMBSI]HSO_4_ (reaction performed with reaction time ranging from 4 to 20 min.), the use of water as the solvent and the energy economy achieved while using sonochemistry as the activating mode turned out to be an attractive improvement. However, ILs exhibit a major drawback for large scale syntheses, namely, their relatively high costs. This is the reason why cheaper catalysts are continuously researched. Lastly, indeno-fused structures have been successfully synthesized by mechanosynthesis while simply using *p*-toluenesulfonic acid (PTSA) as the catalyst [243,244]. One of the most important principles of Green Chemistry consists in the development of environmentally benign synthetic methodologies enabling to avoid the use of solvents, to use environmentally friendly solvents or to reduce the quantity of solvent used. In this field, mechanochemistry is a promising alternative to conventional methods, notably for the synthesis of indeno-fused structures. While using PTSA as the catalyst, the reaction could be successfully performed in solvent-free conditions in a mortar by grinding at room temperature. Reaction yield ranging from 70 to 86% could be determined (see Figure 33, molecules **224**) [243]. PTSA was also used as the catalyst for MCR reactions performed in water at 90 °C for 2.5 h, providing the targeted compounds with higher reaction yields compared to that obtained by mechanosynthesis (see Figure 33, molecules **224**) [244]. Although the reaction yields and the reaction time may be a bit less interesting than that obtained for the different examples presented before, interest of PTSA is that this catalyst need not be prepared, contrarily to the different iron particles or ionic liquids previously mentioned.

A slightly different type of indeno-fused structures deserves to be mentioned in this section, namely the indenopyrimidine derivatives (see Figure 33, molecules **224**) [245]. Even if the central core of pyrimidine contains two nitrogen atoms instead of one for the previous structures, the strategy used to prepare these structures remains the same versus those previously described, consisting in a Knoevenagel reaction activated by the presence of the catalyst, followed by a Michael addition, a cyclization reaction and finally, an aromatization as classically observed for azafluorenones. In 2018, a heterogeneous catalyst, Ag_2_O–ZrO_2_, was proposed to catalyze the MCR, where zirconia was used as the support to immobilize Ag_2_O, which was the key part of this catalyst. Notably, Ag_2_O was capable to coordinate the aldehyde and favor the condensation of indane-1,3-dione **4** (see Figure 41). Recyclability of the catalyst was also quite interesting since a reaction yield of 90% could still be obtained after six cycles (starting from 96% yield for the first run). Although the reaction was performed in ethanol for 30 min, at room temperature, the catalyst only needed to be washed with acetone and dried at 100 °C for 3 h before being reused. Parallel to this, all compounds could be purified by recrystallization in ethanol, evidencing once again the compatibility of the green protocols for the synthesis of complex structures, as exemplified with these indenopyrimidine derivatives.

### 3.10. Indane-1,3-Dione: Versatile Building Block for Spirocyclic Compounds Synthesis

Spirocyclic compounds are chemical structures where two cyclic rings are linked by at least one atom. Such a configuration is strongly present in natural compounds [246]. Spiroindanediones moieties are also present in numerous bioactive compounds [247], and these molecules are efficient as antitumor and antibiotic compounds [248] and antiproliferative molecules [249]. Synthesis of these spiroindanediones can be relatively complex, since the formation of these spiro compounds can led to a wide range of isomers depending on the regioselectivity, the diastereoselectivity and even the enantioselectivity of the reaction. Even if complex mixtures can be obtained during their syntheses, this approach remains however the only one to form elaborated structures. Cycloaddition, domino reaction, and multi-component reactions (MCRs) are the main synthetic pathways leading to spiroindanediones, and these different reactions are described successively. Finally, the other synthetic routes giving access to spiroindanediones are briefly described in the final part (see Figure 42).

#### 3.10.1. Synthesis of Spiroindanediones by Cycloaddition

Spiroindanediones can be synthesized by various types of cycloadditions. Azomethine ylide is a versatile reactant capable to easily reacting with arylidene-1,3-indanedione in 1,3-dipolar cycloadditions or in [3+2] cycloadditions. Azomethine ylide can also be generated by various procedures, such that this reactant was involved in numerous reactions [250].

It was notably used to synthesize spiropyrrolidines starting from 2-ferrocenylidene-2(*H*)-indane-1,3-dione **230** [251]. 2-Ferrocenylidene-2(*H*)-indane-1,3-dione **230** could be synthesized by a Knoevenagel reaction between ferrocene-carboxyaldehyde **229** and indane-1,3-dione **4** (see Figure 43). Then, by mixing sarcosine **232** and indoline-2,3-dione (isatin) **231**, sarcosine **232** could condense with **231** and after decarboxylation, give rise to the azomethine ylide **234**. This ylide can thus react with **230** in a [3+2] cycloaddition furnishing the ferrocenyl spiropyrrolidine adduct **235** (see Figure 44). Using the same strategy, various compounds (**236**, **238**, **240** and **242**) could be synthesized in good yields (higher than 75% yields) by refluxing the methanol solutions for 12 h (see Figure 45).

Single crystal X-ray diffraction analyses performed on different reaction products revealed how this reaction could give the product as a unique regio and stereo isomer. Notably, isatin **231** can react with different amines and give rise to more complex molecules, for example, by reaction with the azomethine ylide resulting from the reaction between ethyl glucinate hydrochloride **243** and dimethyl but-2-ynedioate **244** (see Figure 46). The azomethine ylide **245** thus obtained is capable to react with an arylidene indanedione **246**, as shown in Figure 46. Such a reactivity was notably used to form various dihydro-spiro[indene-2,3′-pyrrolidines] **247** by a one-pot reaction at room temperature and in polar solvent, using triethylamine as the base. The different products could be obtained with reaction yields ranging between 56% and 69% (see Figure 47) [252]. Examination of the single-crystal X-ray diffraction patterns and the 2D NMR spectra revealed that one diastereoisomer was mostly formed. Such a diastereoselectivity was assigned to a steric effect induced by the ester group and occurred during the reaction of the azomethine ylide with the indane-1,3-dione adduct during the concerted cycloaddition (see Figure 48).

Azomethine ylide can also be formed by a decarboxylative condensation of isatin **231** with 1,3-thiazolane-4-carboxylic acid **248** [252]. Such ylides, when formed in situ, can react with various derivatives of 2-arylidene-1,3-indanediones **249** to give bispiro compounds **250** (see Figure 49) that were tested as inhibitors for M. tuberculosis H37Rv. The reaction yields ranged between 60 and 92% depending on the aryl group. This reaction was also regioselective, furnishing only one diastereoisomer.

Azomethine imine can also be synthesized by condensation of the commercially available 3-pyrazolidinone **251** and benzaldehyde **252**, as exemplified with **253** (see Figure 50) [253]. The resulting azomethine imines **254** could react with different arylidene indane-1,3-diones **255** at room temperature using triethylamine as the organocatalyst. Reaction yields ranging between 65 and 98% could be determined, depending on the substituents [254]. The reaction can operate with a good diastereoselectivity (in most cases (>20/1)), (see Table 2, Figure 51).

The reaction is also tolerant to a wide range of substituents, and the electronic effects induced by the substituents were determined as having no influence on the cyclization reaction, the mechanism involving a π-π stacking interaction between the two reactants (see Figure 52).

Iminodiesters can also be used as precursors for the synthesis of azomethine ylides, and an example is given in Figure 53. This synthesis can be base-directed, giving after a [3+2] cycloaddition of **258** with the arylideneindane-1,3-dione **260** and a lactonization reaction, the chromenopyrrolidine **259** [255]. When DMAP was used as the base, the reaction was driven by electronics factors so that chromeno [3,4-*b*]pyrrolidines were obtained in these conditions irrespective of the aryl derivatives (see Figure 54). The substrate was determined as only slightly influencing the reaction yields. Furthermore, when **257** was substituted with fluorine atoms, lower reactions yields were obtained compared to the other substituents (Br, Cl, OMe, Me, NO_2_, …) (see Table 3 and Figure 55).

Conversely, when 1,1,3,3-tetramethylguanidine (TMG) was used as the base, chromeno [3,4-*c*]pyrrolidine **262** was obtained instead of **259,** as the steric factors became predominant during the cyclization reaction (see Figure 54). The reaction products **259** and **262** differ by the intermediate anions **258** and **261** formed by reaction of **257** with DMAP and TMG. During this reaction driven by steric factors, the reaction yield was lower when **257** was substituted with methoxy groups (see Table 4 and Figure 56).

Influence of the catalysts on the regioselectivity of the reaction was not clearly demonstrated. Furthermore, the formation of H-bonds between reactants was suggested as the key element supporting the formation of a unique isomer. When TMG was used as the catalyst, H-bonding interactions were more pronounced than with the other bases. Various chromeno [3,4-*b*]pyrrolidines **259** (see Table 3) and chromeno [3,4-*c*]pyrrolidines **262** (see Table 4) were synthesized using these procedures.

Other 1,3-dipolar cycloadditions were also performed with trifluoropyruvate imines in order to introduce fluorine groups [256]. Starting from ethyl 3,3,3-trifluoropyruvate **263**, formation of the imine **264** led to a dipolarophile, which could be used for cycloaddition reactions (see Figure 57).

After optimization of the reaction conditions, addition of a phase transfer catalyst (benzyltriethylammonium bromide (TEBAB)) **266** and at least one equivalent of LiOH as the base could lead to a complete conversion of the reactants **255** and **265** with an excellent diastereoselectivity as **267** (see Figure 58).

This procedure was notably tested on various substrates, and the reaction proved to tolerate a wide range of functional groups such as chloride, nitro, methoxy and fluoride groups, even if in this case, the reaction yield was lower. The reaction tolerates other aromatic groups such as thiophene, furane and naphthalene (see Table 5 and Figure 59).

Tamura cycloaddition is a common cycloaddition reaction occurring with an enolisable anhydride is used. A version where an anhydride **270** can react with an arylidene-indane-1,3-dione **255** was developed that allowed for the development of many spiro-indanedione derivatives **271** [247]. In this procedure, a base was used to promote the cycloaddition between **270** and **255**, and various arylidene-indane-1,3-dione **255** could be converted as spiro-indanedione derivatives **271** (see Table 6 and Figure 60).

Coumarin is a common functional group present in various natural products [257]. In order to obtain new coumarin-based molecules with biological activities, a [3+2] cycloaddition procedure between arylidene-indane-1,3-diones **255** and a coumarin-based 1,3-dipole precursor **272** was developed [258]. Such a cycloaddition could be catalyzed by an organic catalyst, and the best conditions were found while using 10 mol% of 1-phenyl-3-((1*R*)-(6-methoxyquinolin-4-yl)((2*R*,4*S*,5*R*)-5-vinylquinuclidin-2-yl)methyl)thiourea QN-T **273** in dichloromethane at 30 °C. Using these conditions, various substrates could be screened as reactants. Thus, 21 coumarin-indanedione derivatives **274** could be synthesized in moderate to high yields, as shown in Table 7 and Figure 61.

The mechanism of cycloadditions could be determined due to several control reactions. In the mechanism, the coumarin first interacts with the catalyst due to H-bonding. After this initial step, the adduct and the arylidene-indane-1,3-dione can react via a [3+2] cycloaddition concerted step furnishing the spiro compounds (see Figure 62).

Organocatalysis was also used to achieve asymmetric cycloadditions between 2-arylidene-indane-1,3-diones **255** and Morita–Baylis–Hillman carbonates **275** [259]. In this procedure, a thiourea-phosphine organocatalyst (**276**) was used to achieve the synthesis of various cyclopentene spiro-indanedione derivatives **277** (see Table 8 and Figure 63). The mechanism proposed by the authors was the following: the intermediates, once activated by the organocatalyst, can release CO_2_, and ^t^BuOH can react in a cycloaddition reaction with arylidene-indane-1,3-dione **255**. After releasing the organocatalyst **276**, the spiro-compounds **277** can be obtained (see Figure 64).

[3+2] Cycloadditions enabling to prepare spiro-indane-1,3-diones can also be achieved with metal catalysts. Palladium is a common metal capable of catalyzing various cycloaddition reactions. Palladium was notably used as a catalyst in the [3+2] cycloaddition of vinylaziridine **278** and indane-1,3-dione derivatives **255** [260]. Such a procedure used a palladium (0) catalyst and a ligand (**279**) whose structures were carefully selected among a series of eleven ligands investigated. The optimal conditions were found to be 2.5 mol% of the palladium catalyst Pd_2_bda_3_ (*tris*(dibenzylideneacetone)dipalladium(0)), 6.7 mol% of ligand **279**, in THF and at room temperature for two days (see Table 9 and Figure 65). The scope of applicability of this reaction was examined with a wide range of substrates. Even if the reaction tolerated a wide range of substituents attached to the arylidene-indane-1,3-diones **255**, the enantiomeric ratio greatly changed with the substrates, as shown in Table 9 and Figure 65.

When a cyclohexyl group was used instead of an aromatic group to prepare the indane-1,3-dione adducts, no reaction could take place, demonstrating the importance of starting from arylidene-indane-1,3-diones. The proposed mechanism involves the formation of a pi-allyl-palladium complex, obtained by the oxidative addition of Pd (0) to vinylaziridine **278**. Then, after an aza Michael addition, the resulting adducts can form H-bonds with the amide ligand, giving rise to two reversible transition states. Then, an enolate ring closure can occur, and on the basis of steric factors, only one diastereoisomer forms (see Figure 66).

Vinylethylene carbonate **282** can also react with arylidene-indane-1,3-diones **255** in [3+2] cycloadditions using a palladium catalyst to give tetrahydrofuran-fused spirocyclic 1,3-indandiones **284** and **285** [261]. Such cycloaddition involves the formation of carbon dioxide and proceeds in the optimized conditions, with a phosphoramidite ligand **L 283**, in chloroform at 0 °C for two days. This reaction could be tested at gram scale. Here again, tolerance of the reaction to the substitution pattern of carbonates **282** and arylidene-indane-1,3-diones **255** was remarkable. A unique product could be obtained in high yield in all cases and with a high enantioselectivity (see Table 10 and Figure 67).

Spirovinylcyclopropaneindanedione (VCP) **287** is a reactive spiro compound, usually obtained by reaction of indanedione **4** with 1,4-dibromobut-2-ene **286** in basic media (see Figure 68) [262].

This spirovinylcyclopropaneindanedione **287** was notably used in a cycloaddition with palladium (0) as the catalyst with a sulfonyl-activated imine **288** [263]. Such a procedure can be advantageously used to increase the size of cycles of spiroindanedione compounds. An example of procedure giving access to the five-membered spiroindanedione **289** in 96% yield and with a diastereomeric ratio of 22:1 is presented in Figure 69.

Spirovinylcyclopropaneindanedione (VCP) **287** is also capable of reacting with enals, as exemplified with cinnamaldehyde **290**, enabling to increase by a [3+2] cycloaddition the size of the cycle of the spiro derivatives **292** from three to five carbons (see Figure 70) [264].

The scope of application of this cycloaddition was rapidly studied, showing that aryl derivatives (**293**) or naphthyl-based enals (**294**) were compatible with this reaction, giving the cycloadducts **295** and **296** in 98% and 92% yields, respectively (see Figure 71).

Spirovinylcyclopropaneindanedione (VCP) **287** can also react with nitroalkenes **297** in cycloaddition reactions, producing spiro-cyclopentane-indane-1,3-diones **298**. Such a reaction typically operates according to a two-step procedure [262]. In a first step, the palladium will oxidatively add to VCP **287**, opening the cyclopropane ring and giving an anion and a π-allylpalladium complex. The anion formed by ring opening is thus sufficiently nucleophile to add on the alkene **297**. The nitro substituent present on nitroalkenes **297** is capable of stabilizing the carbanion, and this anion can give rise to an intramolecular cyclization, regenerating the catalyst (see Figure 72, **298**).

Such a procedure was tested with various nitroalkene derivatives, and the substitution pattern of the aryl substituents (presence of electron-donating or -withdrawing groups) did not impact the reaction process. The different products could be obtained in good yields (>80%). Only when the substituent was *o*-CF_3_C_6_H_4_, the product was obtained with the lowest reaction yield of the series (75%) and with the worse diastereoselectivity (14:1). Even with other aromatic, heterocycle rings and alkane groups, the reaction yields remained good as well as the diastereoselectivity ratio and the enantiomeric excess (see Table 11 and Figure 73).

[3+2] Cycloaddition can also be used to construct oxindole-fused spiropyrazolidine compounds **302** starting from spirovinylcyclopropaneindanedione **287** (VCP) catalyzed by palladium (0) [265]. 1-Benzyl-3-diazoindolin-2-one **300** could react with **289** in the presence of Pd_2_dba_3_ using a chiral (P,N) ligand **301** in toluene at 0 °C, affording various spiropyrazolidines **302** with reaction yields ranging from 39% to 99% (see Table 12 and Figure 74).

The mechanism of cyclization was similar to the mechanism previously depicted for the cycloaddition of VCP **289** with nitroalkenes **297**. Thus, the palladium catalyst opens the cyclopropane ring, giving a zwitterionic species with an anion and a π-allylpalladium complex. Then, a [3+2] cycloaddition can occur between the anion and the diazo group, generating the cyclopentane ring after cyclization (see Figure 75).

Palladium was also used in cooperation with an organic base to promote the annulation of vinylcyclopropanes indane-1,3-dione **287** with *para*-quinone methides **303** via a [3+2] cycloaddition [266]. Association of 1,8-diazabicyclo [5.4.0]undec-7-ene (DBU), a thiourea-based ligand **304**, and Pd_2_(dba)_3_ could catalyze the intramolecular annulation reaction between *para*-quinone methides **303** and 2-vinylspiro[cyclopropane-1,2′-indene]-1′,3′-dione **287** giving the product **305** in 61% yield with a diastereoisomeric ratio of 93:7 (see Figure 76).

Palladium catalysis was also used in synergy with diphenylphosphoric acid **306** [267]. Such a mixture could catalyze the reaction between VCP **287** and various imines **307**, producing **308** (see Table 13 and Figure 77). The different imines used in these reactions were prepared in situ.

Other metals can also be used for the cycloaddition reactions. Thus, copper was notably used to catalyze the [3+2] cycloaddition of 2-arylideneindane-1,3-diones **255** with ketoxime acetates **309** [268]. In a first study, the authors optimized the reaction conditions by screening various copper catalysts such as CuI, CuCN, CuOAc and even solid copper with various additives, at various temperatures and in various solvents. The best conditions to produce **310** were determined as being CuCN as the catalyst while using NaHSO_3_ as the base and dichloroethane as the solvent at 100 °C. Then, a screening of different arylidene-indane-1,3-diones **255** was realized, as shown in Table 14 and Figure 78.

The screening showed that the reaction could work with various aryl derivatives, giving the different products with reaction yields higher than 80%. When the methoxy group was attached in *para*-position of the aromatic ring, the reaction yield was lower than with the other groups. However, when the substituent was another aromatic group such as naphthyl, furyl or thienyl groups, the reaction yield decreased slightly, giving the products in around 50% yield. When the substituent was an alkyl group, such as an isopropyl group, the yield was relatively low (21% yield), demonstrating the importance of an aryl ring on the ketoxime acetates to obtain high reaction yields. The reaction mechanism was investigated by the authors, and the following one was proposed. Thus, after an oxidative addition of the copper catalyst **311** on the oxime **309**, a copper enamide **312** formed. Then, this enamide **312** could undergo an intramolecular annulation with **255**, releasing acetic acid as a by-product and producing **313**. Then, the metallocycle could undergo a reductive elimination, giving the final product **310** and regenerating the catalyst **311** (see Figure 79).

Cobalt (II) was also used as a metal catalyst to initiate 1,3-dipolar cycloadditions between azomethine ylides **314** and 2-arylidene-indane-1,3-diones **255** [269]. This metal cation can be chelated with two phenylalanine units, forming a planar complex of cobalt (II). After a careful screening of the reaction conditions, cesium carbonate was determined as being the best base to produce **315**, and the optimal reaction conditions were determined as being 10 mol% cobalt(*L*-phenylalanine)_2_, 10 mol% cesium carbonate in dichloromethane at −5 °C for two days. Using these conditions, a wide range of azomethine ylides **314** and arylidene-indane-1,3-diones **255** could be tested in these conditions (see Table 15 and Figure 80). The reaction tolerates all substituents examined, except that higher reaction yields were obtained while introducing electron withdrawing groups on the azomethine ylides **314**.

A reaction mechanism was also proposed to explain the stereoselectivity of the reactions. In this mechanism, the azomethine ylides **314** chelate to the cobalt metal, leading to an octahedral complex. This complex will hide the Re face of the azomethine ylides, allowing arylidene-indane-1,3-diones **255** to attack the deprotonated azomethine ylides by the Si face exclusively. This stereoselectivity is directly related to the fact that the two aromatic rings (one of L-phenylalanine and the other on the azomethine ylides) will shield the Re face of the azomethine ylides **255** (see Figure 81).

Ruthenium complexes were also used in combination with visible light to catalyze the [3+2] cycloaddition of 2-arylidene-indane-1,3-diones **255** with methyl 2-(3,4-dihydroisoquinolin-2-yl)acetate **316** [270]. The visible-light catalyzed reaction was tested with various methyl 2-(3,4-dihydroisoquinolin-2-yl)acetate **316**, the substituent varying from ester to nitrile groups on the dihydroisoquinoline core and by testing various groups attached to the aromatic ring of 2-arylidene-indane-1,3-diones **255** (see Table 16 and Figure 82). The nature of the aromatic group on the 2-arylidene-indane-1,3-diones **255** strongly influenced the structure of the final product **317**. Thus, when the aromatic ring was substituted with electron-withdrawing groups, the reaction gave spiro[indene-2,1′-pyrrolo [2,1-*a*]isoquinoline]s **317**, whereas when the substituent on the aromatic ring was an electron-releasing group, 3′-arylspiro[indene-2,2′-oxirane]-1,3-diones **318** were obtained.

Such a difference of products (**317** or **318**) can be explained due to the two plausible mechanisms proposed by the authors and that are depicted in Figure 83 where an azomethine ylide is produced due to the photoredox catalyst. If the double bond of 2-arylideneindane-1,3-dione **255** is electronically depleted due to the electron withdrawing group and the ester group is not too big in terms of size, the azomethine can react with 2-arylideneindane-1,3-dione **255** via a cycloaddition reaction. Conversely, if the double bond is electronically enriched by electron-donating groups and the ester group is bulky (for instance with *tert*-butyl groups), then only the epoxidation reaction will occur.

This difference of reactivity can be advantageously used to produce various 3′-arylspiro[indene-2,2′-oxirane]-1,3-diones **318** (see Table 17 and Figure 84).

[3+2] Cycloaddition is not the only possible cycloaddition capable of produce spiroindanediones. The [2+2+2] cycloaddition is a smart and inventive procedure to construct aromatic rings, or heterocycles. These reactions are often metal-catalyzed [271]. 2,2-Di-2-propynyl-1,3-indandione **320** can be synthesized from indane-1,3-dione **4** and propargyl bromine **319** (see Figure 85), and the resulting adduct **320** is a common substrate used in numerous cycloaddition reactions.

Using Molybdenum hexacarbonyl, it is also possible to realize cyclotrimerization. By reaction of 2,2-di-2-propynyl-1,3-indanedione **320** with propargyl bromide **319** upon catalysis with Mo(CO)_6_, a complex mixture of products was obtained, showing the strong reactivity of the alkyne [272]. If the desired product **321** was obtained in 34% yield, the self-dimerized product **323** was also produced as well as another side-product **322** whose origin was determined due to the elucidation of the mechanism (see Figure 86).

After further investigations, conventional heating was replaced by microwaves heating, giving a better selectivity for the product, and at the same time, use of acetonitrile as the solvent could increase the reaction yield. The scope of application of the reaction was also studied, and various spiroindanediones **325** were obtained by [2+2+2] cycloadditions of **320** and **324** (see Table 18 and Figure 87).

The team of Ratovelomanana-Vidal and coworkers used another metal, namely ruthenium, to catalyze [2+2+2]cycloadditions, enabling to form pyridines [273,274,275,276,277]. Notably, ruthenium complexes were used to undergo a cycloaddition reaction of 2,2-di-2-propynyl-1,3-indanedione **320** with cyanamide **326** (see Figure 88). This reaction could give the expected spiro-compound **327** in 97% yield within 5 min [277]. A similar procedure could be used for another cyanamide, i.e., pyrrolidine-1-carbonitrile **328** (see Figure 89), and the reaction conditions could be greatly improved while using microwave irradiation (see Figure 90). Using these improved conditions, all compounds (**333** and **334**) could be obtained in high to almost quantitative yields [274,276].

The procedure described before used Cp*Ru(CH_3_CN)_3_PF_6_ as the catalyst. However, another procedure was also proposed using RuCl_3_ as a more accessible catalyst and allowing the formation of the spiro compound **335** using a cheaper synthetic approach (see Figure 91) [275]

Selenocyanates such as **336** and **337** could also be used as reactants with 2,2-di-2-propynyl-1,3-indandione **320** to form the spiro compounds **338** and **339** according to the procedure shown in Figure 92 [273].

Other metals were also employed as exemplified with cobalt, which can be used to catalyze the [2+2+2] cycloaddition of 2,2-di-2-propynyl-1,3-indanedione **340** with a methyl indole derivative, namely **341** (see Figure 93) [278]. Rhodium catalysis is also interesting, and in this case, an amide group can act as a directing group to ensure the cycloaddition and allow for the synthesis of polycyclic molecules bearing indanedione moieties such as **344** (see Figure 94) [279].

The mechanisms involved in these two types of catalyzed cycloaddition reactions are similar. The C-H activation allows for the formation of a carbon-metal bond with the indole compound, and after coordination of the diyne, a migratory insertion of the two alkynes gave a metallocycle, which furnished the product after reductive elimination (see Figure 95).

Phosphabenzenes are heterocycles containing one phosphorous atom in the cycle, and such heterocycles are only poorly described in the literature. The synthesis of these molecules involve multi-step reactions, cycloadditions and reversible cycloadditions sequences [280], with dangerous sylilated compounds, constituting a major impediment for their developments [281]. To address this issue, iron was notably used to perform the synthesis of phosphabenzene **346** in safe conditions, involving in one reaction, a diyne **320**, a phosphaalkyne **345** and iron diiodide in xylene (see Figure 96) [282].

However, spiro-indanedione can also be obtained by other cycloaddition reactions that are also metal-catalyzed. For example, by starting from a cyclic carbonate **347** and by using the same procedure of that reported by Guo, various derivatives of **349** could be prepared [261]. Cyclic carbonate **347** can react with various 2-arylidene-indane-1,3-dione **255** in palladium-catalyzed cycloadditions and give six-membered spiro compounds **349** (see Table 19 and Figure 97).

Nickel can catalyze the formation of spiro compounds containing an eight-membered ring cycle [283]. Due to the association of nickel with a NHC ligand **351**, such spiro cyclization of 2,2-di-2-butyn-1-yl-1*H*-indene-1,3(2*H*)-dione **332** and 1-benzhydrylazetidin-3-one **350** could be achieved in toluene after 8 h at 0 °C, furnishing **352** in 88% yield (see Figure 98).

A base-catalyzed [4+1] cycloaddition was also described in the literature, enabling the synthesis of spiroindanes **354** bearing a *para*-phenol moiety (see Figure 99) [260]. Rhodium was also used in metal-catalyzed intramolecular [4+3] cycloadditions of dienyltriazoles **355** to give the spiro-indanedione compound **356** (see Figure 100).

*o*-Quinodimethanes (*o*-QDM) can be used to generate various compounds containing spiroindanediones moieties. Such molecules were obtained through a [4+4] cycloaddition, producing dibenzocycloooctadiene structures (see Figure 101) [284]. Starting from indanone containing benzo[c]oxepines such as compounds **357** and **361**, and by heating in a polar solvent, it was possible to create in situ o-QDM **358** and **362**, producing after [4+4] cycloaddition the different spiroindanediones **359**, **360**, **363** and **364**.

Asymmetric cross [10+2] cycloadditions were also successfully achieved by opposing electron-deficient alkenes such as **365** and 2-arylidene-indane-1,3-dione **249** by phase transfer catalysis [285]. Such reactions can furnish spirofused polycyclic structures such as **367** as shown in Figure 102.

To conclude, cycloaddition reactions can lead to a large variety of spiro-indanediones, even if [3+2] and [2+2+2] cycloadditions are the two privileged routes to synthesize spiro-indanedione moieties. Other cycloaddition reactions can give promising results as shown in this paragraph.

#### 3.10.2. Synthesis of Spiro-Indane-1,3-Diones by Domino Reaction

Domino reactions are defined as chemical processes where the final product comes from a sequence of reactions. The product of a first reaction become the reactant of another one. In contrast to a one-pot procedure or a multi-component procedure, the reaction conditions cannot be changed after the beginning of the reaction, and no additional compounds can be introduced in the reaction mixture.

One of the first examples of domino reaction described in the literature concerned the synthesis of spiro-compounds **378** by means of a domino Knoevenagel/Diels–Alder /Epimerization sequence [286]. This domino reaction was performed at room temperature in methanol for four days (see Table 20 and Figure 103).

The mechanism supporting the chemical structure of the final products was suggesting as proceeding according to the following steps (see Figure 104): First, the amine, i.e., L-proline, catalyzes the classical Knoevenagel condensation between **4** and **368**, providing **255**. In a second step, L-proline reacts with the Michael acceptor **369**, generating the diene **370**. Then, the diene **370** can react with the dienophile **255**, regenerating the amine. Finally, by epimerization still in the presence of L-proline, the final product **370** can be obtained.

This strategy was notably applied to the construction of chiral spiro[indane-1,3-dione-tetrahydrothiophenes] **373**. For this reaction, a tertiary amine-thiourea organocatalyst **372** was used, enabling a sulfa Michael/Michael sequence to occur [287]. By mixing indane-1,3-dione **4**, an aldehyde **368** and a thiol, the aforementioned organocatalyst **372** and molecular sieves in toluene at −20 °C, spiro-compounds could be obtained. The scope of application of this reaction was examined with various aromatic groups (see Table 21 and Figure 105).

X-ray structure of one of the substrates could be obtained and enabled to determine how the organocatalyst was interacting with the substrate and could promote the reaction at the Si face of the substrate (see Figure 2).

More recently, the scope of application has been expanded to the synthesis of spiro tetrahydrothiophene-indan-1,3-diones **376** starting from 1,4-dithiane-2,5-diol **374**. The reaction was performed at room temperature in dichloromethane [288]. The scope of application of this reaction was studied, as shown in Table 22 and Figure 106.

Lewis acids were also used to synthesize spiro compounds by domino reactions, and the combination of a Knoevenagel condensation and a 1,3-dipolar cycloaddition was notably examined. ZnCl_2_ was used as the Lewis acid-based catalyst and bromonitrile oxide **377** as the main reactant. Starting from indane-1,3-dione **4**, aromatic aldehydes **368** and dibromonitrile oxide **377** in basic conditions in THF at 45 °C, spiro-compounds comprising an isoxazole moiety **378** could be obtained (see Table 23 and Figure 107) [289].

The same procedure was also applied to the synthesis of spiro-compounds **380** by replacing the former aromatic aldehyde **368** with benzoimidazoles **379** (see Table 24 and Figure 108), thiazole **382** or benzothiazole **383** (see Figure 109). These two reactions can lead to interesting compounds since the two families of products (**380**, **384** and **385**) were tested as ligands for coupling reactions.

Other domino reactions involved a Michael addition followed by a 1,3-dipolar cycloaddition of 2-arylidene-1,3-indanediones **255** and 5-aryl-1,3,4-oxathiazol-2-ones **388** in toluene (see Figure 110) [290]. A mechanism involving the formation of a benzonitrile sulfide intermediate was proposed by the authors (see Figure 111).

Phosphine can also act as an initiator for the synthesis of spiro compounds containing cyclopentanones [291]. In this aim, 2-arylidene-indane-1,3-diones **255** and ynones **391** were mixed in EtOH. The reaction was studied for different ynones **391** and 2-arylidene-indane-1,3-diones **255** (see Table 25 and Figure 112). Since 2-arylidene-indane-1,3-diones **255** can be synthesized by a Knoevenagel reaction, a tentative of one-pot procedure involving indane-1,3-dione **4**, an aromatic aldehyde **393** and an ynone **394** revealed than the one-pot procedure was possible and that the separated synthesis of 2-arylidene-indane-1,3-diones **255** was not necessary (see Figure 113). Moreover, a mechanism was proposed, where the phosphine binds to the alkyne **394** in first step, creating a α,β-unsaturated ketone **396**. Then, the phosphorous will stabilize the oxygen, allowing for the formation of a carbanion in the methyl in α-position of the ketone (**397**), and the resulting conjugated system **397** can react with 2-arylidene-indane-1,3-diones **255**. The carbanion **398** formed during the addition is stabilized by the two ketones of indane-1,3-dione and can undergo an addition at the C3-position of the α,β unsaturated ketone, generating a cyclopentanedione **399**. The phosphine is then regenerated, providing the product **395** (see Figure 114).

Silver can be used as a catalyst in association with diphenylphosphine oxide **402** and a magnesium salt as an additive to synthesize spiro-compounds **403** [292]. The mechanism of reaction involves the formation of a radical that can undergo a cyclisation process (see Figure 115). Such a reaction was used to synthesize spiro-compounds with indane-1,3-dione (see Figure 116).

Gold was also used as a catalyst for domino reactions. Starting from indane-1,3-dione **4**, a double propargylation of indane-1,3-dione produced **320**, which was converted as **404** by the mono functionalization of one of the two propargyl groups by a Sonogashira reaction. Compound **404** was later used for the gold-catalyzed enediyne cyclization (see Figure 117).

Then, a domino intramolecular cyclization could be carried out with **404** leading to 5′-hydroxy-6′-methyl-1′,3′-dihydro-2,2′-spirobi[indene]-1,3-dione **406** using **405** as the catalyst (see Figure 118).

In 2020, a cascade Michael addition/cycloaddition reaction between 2-arylidene-indane-1,3-dione **255** and allenoates **407** was reported [293]. Various spiro-derivatives **408** could be obtained by this domino reaction, the overall reaction being catalyzed by a phosphine (see Table 26 and Figure 119). The mechanism is the combination of a Michael addition of activated allenoates (A) on 2-arylidene- indane-1,3-diones **255**. After a proton migration, a second 2-arylidene-indane-1,3-dione is attacked, forming (D). Then, by an intramolecular cycloaddition, (E) is formed, and by regeneration of the phosphine catalyst, the final product (**408**) can be obtained. This reaction also led to the by-product **409** coming from the [4+2] cycloaddition of the activated allenoates to the arylidene indanedione, furnishing 4aa (see Figure 120).

A domino process was also reported with 2-arylidene-indane-1,3-diones **255** and *N*-alkoxyacrylamides **410** in the presence of a base and allowing for the formation of spiro-compounds **411** bearing an indane-1,3-dione moiety and a lactam group [294]. In this process, twenty different derivatives were obtained (see Table 27 and Figure 121).

A mechanism was tentatively proposed by the authors, where the base activates first *N*-alkoxyacrylamide **410**, and the resulting anion **412** can thus react with 2-arylidene-indane-1,3-dione **255** in an aza-Michael reaction followed by an intramolecular Michael addition, forming a spiro compound **414** containing an enol. By reaction of this enol **414** with another *N*-alkoxyacrylamide **410**, the final product **411** can be obtained (see Figure 122).

In order to achieve the synthesis of more complex molecules that can exhibit biological properties, a cascade double Michael addition/acetalization proved to be an interesting approach to synthesize complex spiro indane-1,3-dione derivatives **416** and **417**. The reaction between a 2-hydroxyarylidene-indane-1,3-dione **260** and hexenedione derivatives **415** could lead to various spiro compounds **416** and **417**, when the reaction was catalyzed by DABCO (see Table 28 and Figure 123) [295]. A mechanism was notably proposed: firstly, the enolate of the dione **415** is formed by deprotonation of the amine (see Figure 124). Then, the enolate can proceed to the nucleophilic attack onto the Michael acceptor, i.e., 2-arylideneindane-1,3-dione **260** (see Figure 124). Then, the enone part of the hexenedione **261** can act as a second Michael acceptor. The indane-1,3-dione anion can also attack the Michael acceptor. An acetalization reaction can occur when the deprotonated hydroxyl group of indane-1,3-dione attacks the enolisable ketone, and after protonation, DABCO is regenerated, and the product **416** is formed (see Figure 124).

A quadruple cascade reaction was also reported to create complex molecules starting from 2-arylidene-indane-1,3-diones **260** and conjugated enals **421**. An iminium–enamine–iminium–enamine sequential activation followed by an oxo-Michael addition showed that it was possible to synthesize complex molecules bearing indanedione moieties (see Figure 125) [296]. The mechanism proposed by the authors was the following one: after condensation of proline on the α,β-unsaturated aldehyde, the resulting iminium **424** could be attacked on its Re face by the alcohol of 2-arylidene-indane-1,3-dione. This oxo-Michael reaction forms an enamine **425** that can react via an intramolecular Michael addition, giving a nucleophilic iminium **426** capable of reacting with another equivalent of α,β-unsaturated aldehyde. The enamine **427** thus obtained can react in an intramolecular condensation, giving the final product **423** and releasing the catalyst (see Figure 126).

#### 3.10.3. Synthesis of Spiro-Indane-1,3-Diones by MCR

Multicomponent reaction (MCR) is a process where more than two chemical reagents react together to form one product. Such processes are interesting, since they reduce the number of steps to form the final product, facilitate in the purification of the product by avoiding the presence of side-products and, enable the perfect control of the stereoisomeric parameters at the same time.

With regard to the interest of MRC, numerous spiro-indane-1,3-diones **428** were prepared with this procedure. By use of pyridine as the base, spiro-indane-1,3-diones **428** could be easily synthesized starting from indane-1,3-dione **4**, an aromatic aldehyde **429** and a pyridinium ylide **431** [297]. The pyridinium ylide **431** could be synthesized from an aromatic ketone **432** and pyridine, furnishing in a first way the pyridinium salt **433**. Then, this pyridinium salt **433** can be converted as a pyridinium ylide **431** due to a base. Parallel to this, indane-1,3-dione **4** can react with the aromatic aldehyde **429** in a Knoevenagel condensation, furnishing 2-arylidene-indane-1,3-dione **430**. Then, the pyridinium ylide **431** can add on the 2-arylidene-indane-1,3-dione adduct **430**, furnishing in turn **428** (see Figure 127). Due to this procedure, various aldehydes were tested, and moderate to good yields were obtained during the screening of the different aldehydes (see Table 29 and Figure 128).

Benzothiazole can also be used to achieve the synthesis of spiro compounds starting from indane-1,3-dione **4** [298]. By using indane-1,3-dione, dimethyl but-2-ynedioate **434** and substituted benzothiazoles (**435** or **436**), spiro compounds **437** and **438** could be respectively obtained with 2-methylbenzo[d]thiazole **435** and 2,5-dimethylbenzo[d]thiazole **436** (see Figure 129). The mechanism of the reaction was not clearly established, but it seems to proceed via the formation first of a nitrogen-carbon bond between benzothiazole and the alkyne, and in a second step of the formation of a carbon-carbon bond between the alkyne and indane-1,3-dione **4** (see Figure 130).

Synthesis of spiro compounds can also be obtained through a microwave and catalyst-free procedure. By mixing proline **439**, an aromatic aldehyde **368** and 2-arylidene-indane-1,3-diones **255** and by using a microwave-assisted synthesis, spiro-*N*-fused indanedione compounds **441** and **442** could be successfully prepared [299]. In this procedure, the combination of a condensation, a decarboxylation and a 3+2 cycloaddition, could produce two isomers **441** and **442**, as shown in Figure 131. The scope of application of this reaction was studied with various aromatic substrates (see Table 30 and Figure 132).

A one-pot five-component reaction was also developed, associated with a CuAAC (Copper catalyzed alkyne azide cycloaddition), a [3+2] cycloaddition and a condensation, furnishing triazole-containing spiro-indane-1,3-diones **435** [300]. In this procedure, five components were used, namely indane-1,3-dione **4**, an aromatic azide **444**, an aromatic aldehyde **368**, 1-(prop-2-yn-1-yl)indoline-2,3-dione **443** and sarcosine **232** that could react with copper sulfate, sodium ascorbate as the catalyst in PEG 400 as the solvent. This reaction proved to be versatile since various aromatic azides **444** or aromatic aldehydes **368** could be used (see Table 31 and Figure 133). The mechanism proposed by the authors demonstrated that two products can be obtained. However, for unexpected reasons, the reaction proved to be regioselective, and **445** was obtained as the unique product of the reaction.

MCR conditions were also used for the design of spiro-indandiones exhibiting medicinal properties [249]. Indane-1,3-dione **4**, aromatic aldehydes **368** and methyl enones **446** were mixed together with an organocatalyst **447**, allowing for the formation of spiro compounds **448** containing a halogenated aromatic ring capable of reacting subsequently in a Suzuki cross-coupling reaction, and allowing for the formation of various derivatives **450**, which were tested for biological applications (see Table 32 and Figure 134).

Nanoparticles were also used as catalysts in one-pot three-component reactions [301]. By using proline-functionalized Fe_3_O_4_ particles (LPSF) and DABCO as the base, spiro-cyclopropanes **452** could be synthesized starting from indane-1,3-dione **4**, aromatic aldehydes **368** and aromatic ketones **451**, bearing a bromine in β-position. Various derivatives were prepared due to this strategy (see Table 33 and Figure 135). Another advantage of this strategy is that iron-based nanoparticles were magnetic, allowing for an easy recovery of the catalyst. The mechanism was supposed to work as depicted below: the nanoparticles due to the amine groups present on the external core of the nanoparticles can interact with substituted benzaldehyde **368** and indane-1,3-dione **4**, promoting the Knoevenagel condensation and providing 2-arylidene-indane-1,3-dione **255**. Parallel to this, the ketone **451** can interact with nanoparticles (interaction between the amine and the ketone), and then DABCO can easily react with the bromine atom, forming a zwitterion, composed of the quaternary amine and a carbanion in beta position of the ketone. The zwitterion thus prepared can undergo an addition onto 2-arylidene-indane-1,3-dione **255**. Then, an intramolecular cycloaddition reaction can give the desired product **452** (see Figure 136).

If cycloadditions, Domino processes and multicomponent reactions can lead to numerous spiro compounds; several other procedures were also developed to access to spiro compounds.

#### 3.10.4. Synthesis of Spiro-Indane-1,3-Diones by Miscellaneous Way

In this part, all the other synthetic procedures leading to spiro-compounds are discussed. pH-switchable compounds belong to a recent concept that is born with molecular machines [302]. Using 5-(2-bromoethyl)phenanthridin-5-ium bromide **453** and indane-1,3-dione **4**, a simple reaction in basic media could lead to the formation of a spiro compound, where the spiro compound can be in a closed/opened position **454a/454b** depending on the pH (see Figure 137). Such pH switchable compounds have the advantage to be easily tunable in terms of absorption and emission maxima [303].

Indane-1,3-dione **4** is an interesting scaffold that was extensively used in organic photovoltaics cells. In this way, it is interesting to synthesize new molecules for optoelectronic applications. Synthesis of spirobisindanedione was successfully obtained through a multi-step synthesis. The same team succeeded in synthesizing bindone **50** as well as spiroindanedione **455-457** (see Figure 138) [40]. Indane-1,3-dione **4** by reacting with phthaloyl dichloride **458** and sodium hydride, and depending on the reaction conditions used, can give **50** or **456** (see Figure 139). The reaction of oxidative cleavage of **456** with sodium periodate and ruthenium trichloride can give the spirobisindanedione **455** (see Figure 140). A similar version of these molecules was also performed with a bromine atom attached to indane-1,3-dione (see Figure 141).

Iron can be used as a catalyst in an alkene hydrofunctionnalization with 2-arylidene-indane-1,3-diones **255** [304]. In this process, an iron (III) catalyst, the selected 2-arylidene-indane-1,3-dione **255** could react with a terminal bromo-alkene **462**. Various substrates were tested (see Table 34 and Figure 142), and a mechanism was proposed by the authors to support the synthesis of **463** (see Figure 143).

Copper was also used as a catalyst for the synthesis of spiro compounds. In a [3+2] radical cycloaddition catalyzed by copper, 2-arylidene-indane-1,3-dione **255** could react in the presence of *N*-acetyl enamides **464** at high temperature [305]. Many *N*-acetyl enamides **464** and 2-arylidene-indane-1,3-diones **255** were mixed, providing a wide range of spiro structures **465** (see Table 35 and Figure 144). The plausible mechanism involves an air oxidation of the copper (I) complex to a copper (II) complex, producing in the meantime a radical (see Figure 145).

Copper was also used in combination with oximes to produce spiro indanedione derivatives **467** (see Table 36 and Figure 146) [306]. A copper (I) salt was used as the catalyst, and the mechanism was similar to the previous mechanism, in which the copper catalyst is first oxidized, allowing for the formation of a radical. Then, the copper complex forms a metallocycle, where the oxidation state of copper is Cu(III). Being unstable and by reductive elimination, the product **467** can be formed and the copper catalyst regenerated (see Figure 147).

Seyferth–Gilbert reagent **468** is a reagent classically used to transform carbonyl groups into alkynes. Such a reactant was notably used in association with indanedione derivatives **255** to form spiropyrazolineindane-1,3-diones **458** using CsF as the base [307]. When the 2-arylidene-indane-1,3-diones **255** and the Seyferth–Gilbert reagent **468** were mixed with sodium hydroxide, 3-pyrazolylphthalide **470** could be obtained. Therefore, the possibility to design various structures by modifying the quantities and the nature of the base was demonstrated, as shown in Figure 148. By using CsF in 0.1 equivalent, the authors could synthesize a wide range of spiropyrazolineindane-1,3-diones **469** (see Table 37 and Figure 149).

To sum up, spiroindane-1,3-dione are interesting structures that can be obtained using various synthetic procedures. Due to their similitudes with natural compounds, these compounds are promising structures for the design of biologically active compounds, active pharmaceutical ingredients or biomimetic compounds. If cycloaddition remains one of the privileged ways to produce spiroindane-1,3-diones, highly complex structures could also be prepared and imply the development of appropriate synthetic procedures such as domino or multicomponent reactions. The list of the synthetic strategies developed to access to spiro compounds has been exhaustively detailed. Furthermore, other reviews were written on these topics, specially devoted to the synthesis of spiro compounds [308]. The importance of such moieties shows that this field is clearly not unveiled, and future discovery will allow for the design of highly biologically active spiroindane-1,3-diones.

## 4. Applications of Indane-1,3-Dione-Based Structures

### 4.1. Photopolymerization

During the past decades, substantial efforts have been devoted to develop photopolymerization processes under visible light and low light intensity [309,310,311,312,313,314,315]. Visible light is a safe spectral range of irradiation for the manipulators, and light is also a traceless reagent, making photopolymerization a green approach for the design of polymeric materials. Several parameters govern the photoinitiating ability of the photosensitizers such as their molar extinction coefficients, their redox properties and notably their easiness to be oxidized or reduced, depending of the additives used in the photoinitiating system [316,317,318,319,320,321,322,323,324,325]. Additionally, efficiency of the polymerization process is also highly dependent on the rate constant of interaction with the additives [326]. With the aim at developing dyes with high molar extinction coefficients, push–pull dyes are the most favorable structures, as the careful selection of the electron-donating and electron-accepting moieties connected at both ends of the π-conjugated spacer can efficiently tune the broadness but also the position of the intramolecular charge transfer band [327,328,329,330]. In this field, indane-1,3-dione and its derivatives have been extensively studied as photoinitiators, and a selection of structures (**470** [331], **471–475** [332], **476** [333], **477** [334], **478–497** [335]) is presented in Figure 150. Among the most interesting findings, photoinitiating ability was demonstrated as being directly related to the solvatochromic properties of the push–pull dyes. Only dyes for which linear correlations using empirical solvent polarity scales (Bakhshiev’s [336], Kawski−Chamma−Viallet’s [337], Lippert−Mataga [338], McRae’s [339], and Suppan’s [340] solvatochromic scales) could be established to initiate a polymerization process. If the direct relation existing between solvatochromic properties and photoinitiating abilities could be demonstrated, no clear explanations could be provided to support this unexpected behavior. Intrigued by these results and considering that little exploration as to the scope of push–pull dyes as photoinitiators has been disclosed, in 2020, Lalevée and coworkers examined this point with a new series of 21 dyes **478–497** in which the optical properties were finely tuned by modification of the electron-accepting core through an extended π-conjugation or by converting **4** and **68** into stronger electron acceptors.

All dyes showed an excellent ability to initiate the free radical polymerization of acrylate (Ebecryl 40) upon irradiation with a light-emitting diode (LED) at 405 nm, which is the wavelength currently under use for 3D printers [341,342,343]. As an appealing feature, beyond simply initiating a polymerization process, dyes **486** could also exhibit excellent photobleaching properties, what is rarely observed and what is actively researched for visible light photoinitiators, considering that these dyes are highly colored compounds often imposing their own colors to the final coating [344,345,346].

### 4.2. Non-Linear Optical Properties

Push–pull dyes exhibiting a large ground state dipole moment usually have high molecular non-linear optical (NLO) efficiencies [347,348]. Considering this, indanedione-1,3-dione **4**, by its electron-withdrawing ability, is an excellent candidate for the design of dyes exhibiting such a property [349,350]. To present large optical nonlinearities, molecules should be organized in a none-centrosymmetric arrangement, and different strategies have been developed to stabilizing the dipole orientation. Thus, introduction of indane-1,3-dione derivatives into polymer composites (**498–500**) [351] or formation of Langmuir-Blodgett films with amphiphilic derivatives (**501–504**) [352,353] proved to be effective strategies to address this issue (see Figure 115). In this last case, introduction of an additional double bond did not significantly modify the structure of the Langmuir–Blodgett films. Highest hyperpolarizability was obtained with **501**, indicating the high order of the LB film obtained with this molecule. Parallel to this, an increase in the number of layers enhanced the NLO signal. Covalent linkages to polymers proved as being another strategy to retain the molecular orientation obtained by electrical poling [354]. By heating the polymer **505** at a temperature higher than its T_g_ and upon application of an intense electric field, an orientation of the push–pull dyes connected to the polymer backbone could be obtained. While maintaining the polymer film at a temperature lower than the polymer T_g_, relaxation of the chromophore alignment could be efficiently slowed down, maintaining the molecular orientation of the dyes over time. Improvement of the molecular hyperpolarizability could also be obtained by improving the electron withdrawing of the acceptor, as exemplified with the dye **506** based on 2-methylidene-3-(dicyano-methylidene)-1-indanone [24]. Comparison with a reference compound, i.e., disperse red 1, revealed 243 to exhibit an hyperpolarizability as high as 1558 × 10^−30^ esu·D, greatly higher than that of the reference compound (814 × 10^−30^ esu·D). Most of the dyes prepared for NLO applications are synthesized by means of a Knoevenagel reaction, as exemplified by the selection of molecules **498–506** presented in Figure 151.

### 4.3. Fluorescent Chemosensors and Chemodosimeters

The detection of metal cations, halide ions, cyanides and even of neutral species has been an active research field so that two types of detectors were developed [355]. The first category of fluorescent sensors are those comprising a binding site giving rise to an irreversible chemical reaction with ions, and these first types of compounds are named chemodosimeters. The second type of optical sensors are those capable of initiate a communication mechanism between the binding site and the ions, but in a reversible way. In this last case, these compounds are thus named fluorescent chemosensors. Indane-1,3-dione **4** has been at the origin of the elaboration of an efficient chemodosimeter for cyanide detection. Cyanides are extremely toxic anions so that a concentration as low as 2.7 µM is tolerated in drinking water [356,357]. In 2009, a chemodosimeter based on calix[4]pyrrole with an indane-1,3-dione unit at the β-pyrrolic position was reported by Lee and coworkers [213]. A dependence of the dye discoloration with the cyanide concentration was clearly evidenced, and a disappearance of the yellow color of **507** upon addition of cyanides could be easily detected with the naked eye. Chemodosimeter proved also to be highly selective since only cyanide ions were detected even when hidden within other ions. The remarkable selectivity and affinity for cyanide anions was explained by the fast equilibrium process involved in the complexation followed by the reaction induced by this extremely nucleophilic anion. After complexation of cyanide anions in the binding site of the calix[4]pyrrole, a nucleophilic addition at the β-position of the indane-1-3-dione group could occur, as shown in Figure 152.

Based on the same nucleophilic addition of cyanides onto the double bond of push–pull dyes developed with indane-1,3-dione, **508** could be used as a dosimeter enabling to combine two detection modes [358]. Thus, upon addition of cyanide anions on **508**, a clear discoloration of the dye could be detected with the naked eye. Parallel to this, a complete quenching of luminescence could be evidenced by photoluminescence measurements. If the detection of cyanides anions in THF was extremely fast, in water, kinetic of addition was considerably reduced, assigned to a solvatation effect of water molecules around the cyanide anions (see Figure 153). Here again, **508** proved to be selective for the detection of cyanide anion, and this is again related to the small size of cyanides but also to their averred nucleophilic character. A similar strategy was developed with diketopyrrolopyrrole **509** [359]. A dual mode of detection was also observed with this compound. In this last case, a mechanism of exciplex formation was proposed to support the fluorescence quenching observed experimentally (see Figure 153 and Figure 3).

### 4.4. Solar Cells

Organic solar cells have received significant attention for the possibility to convert photons to electrons. Among the most widely studied electron-acceptors, [6,6]-phenyl C_61_ butyric acid methyl ester (PC_61_BM) is the most popular one [99,360,361,362]. Due to a severe phase segregation upon aging of the device, the low solubility of PC_61_BM in most of the common organic solvents and non-fullerene acceptors has been identified as promising alternatives to address the segregation issue and the solubility issue. Furthermore, the photon-to-electron conversions remain low, lingering around 3–6% for non-fullerene acceptors [363,364,365,366,367,368,369,370,371,372,373,374,375,376,377,378], justifying the constant efforts to develop new structures. To create strong electron acceptors that could replace PC_61_BM, indane-1,3-dione **4** was notably combined with dibenzosilole [379]. A photon-to-current conversion efficiency (PCE) of 2.76% could be obtained for the solution-processed devices comprising a poly(3-hexylthiophene) (P3HT)/**510** blend as the active layer. By replacing dibenzosilole by a naphthalimide unit, encouraging results were obtained with **511**, with a PCE reaching 3.52% in the same conditions [380]. However, other authors also tested the opposite situation, using the dibenzosilole-based compounds as electron donors [381]. End-groups on **512** and **513** were determined as drastically affecting the morphology of the active layer, more than the photophysical properties of the dyes. Even after solvent vapor annealing, the power conversion efficiency of solar cells fabricated with the **512**/PC_61_BM blend remained low, peaking around 0.5% contrarily to 6.6% for devices fabricated with the **513**/PC_61_BM blend. The excellent electron-donating ability of **513** can also be assigned to the presence of the thiophene moiety, reported as improving both the optical and photovoltaic properties of the dyes comprising this group [382]. This trend was confirmed with **514** [11] and **515a** [383] or **515b** [384], with which a PCE of 2.4%, 6.46% and 8.22% were determined with solar cells of similar structures than that used for **513**. In the case of **515b**, improvement of the photovoltaic properties was assigned to the more balanced charge transportation in the devices and the use of copper isothiocyanate acting as a hole transport/injection layer. However, counterexamples exist, as exemplified with the asymmetric structure **516** [385]. Photovoltaic properties of **516** in bulk heterojunction solar cells remained poor, and a PCE of 1.7% was obtained for an active layer composed of a 1:1 **515**/PC_61_BM blend. This low efficiency was notably assigned to the low hole mobility of **516** and adverse charge transport within the active layer (see Figure 154). Finally, a similar low power conversion efficiency was obtained with **517** and **518**, still based on an asymmetric structure [386]. Here again, the low efficiency (0.24–0.33%) obtained with **517** and **518** was assigned to an unfavorable morphology of the active layer as well as its high roughness.

## 5. Biological Applications

Indane-1,3-dione **4** has been the focus of intense research efforts in medicinal chemistry since the demonstration in the early 1930s of the bacteriostatic activity of indane-1,3-dione and related derivatives, but also of many 1,3-diketo compounds showing interesting physiological activities [387,388]. This section provides an explicit overview of the importance of indane-1,3-dione **4** as a building block for the design of molecules with potential biological applications.

### 5.1. Indane-1,3-Dione as Antimicrobial Agent

Any agent that kills or slows down the growth of a micro-organism may be defined as an antimicrobial agent. The increasing demand of antimicrobial drugs resulting from a faster microbial resistance to drugs requires the development of new compounds of innovative structures. Thus, antiseptics have been designed with indane-1,3-dione **4** as soon as 1931 by Walker et al. and their biological activities improved one year later by Robinson et al., who developed 1,3-diketo systems exhibiting remarkable physiological properties In Robinson’s work, the indane-1,3-dione core was formed by a Friedel–Craft reaction, enabling to introduce various *n*-alkyl groups or hydroxyl groups onto the aromatic core (compounds **521**) (see Figure 155). Notably, the presence of phenol groups was identified as improving the bacteriostatic effect of the resulting compounds [388,389].

All of these compounds were tested in vitro on Gram-positive bacteria: *Staphylococcus albus* Rosenbach (*S. epidermidis*), Staphylococcus aureus (*S. aureus*), (*Bacillus megatherium* (*B. megatherium*), *Bacillus subtilis* (*B. subtilis*), *Bacillus mycoides* (*B. mycoides*), Gram-negative bacteria: *Bacterium pyocyaneum* (*P. aeruginosa*), *Bacterium prodigiosum* (*B. Prodigiosum*) and *Escherichia coli* (*E coli*) but also on acid-fast bacteria such as *Mycobacterium phlei* (*M. phlei*) according to the Rideal–Walker method. Even though this method is not used anymore due to its lack of reliability on phenols, it was replaced by the McFarland protocol to furnish more reproducible data and to determine the minimum inhibitory concentration (MIC) [390]. This difference of protocols does not facilitate the comparison with recent research, but the activity of other phenol compounds still allows us to obtain a general picture of the antiseptic properties of this set of compounds. Notably, Robinson’s team noticed that all the molecules they developed were particularly effective against Gram-positive bacteria but had small efficiency against Gram-negative bacteria. Concerning the acid-fast bacteria (*M. phlei*), some molecules could also kill these organisms but only at high concentrations. Finally, the authors demonstrated an improvement of the bactericidal activity by increasing the number of carbons in the side-chains toward Gram-positive bacteria but also on Bacterium typhosum, responsible for typhoid fever [391]. This effect reaches its maxima for three molecules (see Figure 156: **522**, **523** and **524**), with an activity more than ten times higher than that of the methyl or p-n-octylphenol (expressed in equimolecular phenol coefficient of bactericidal power). Even if these molecules have not been used as antibiotics later, modern researchers may inspire for sure from these former studies to design modern drugs due to the rise in antibiotic resistance.

The place of indane-1,3-dione in this field has clearly evolved. A structure related to indane-1,3-dione, namely indan-1-one, also showed interesting antimicrobial properties so that these two structures were often studied concomitantly for the design of antimicrobial compounds [1,2,3]. Considering that the biological activity of indane-1,3-dione **4** can be greatly improved by chemical engineering, indane-1,3-dione **4** has thus been extensively used as a building block in multicomponent reactions (MCRs) to combine the properties of indanones or indane-1,3-dione with that of other structures also displaying physiological properties.

A relevant example of this strategy has been reported in 2011 with a study devoted to the antimicrobial activity of 1,3-disubstituted indeno [1,2-*c*]pyrazoles [392]. Pyrazoles are an important class of pharmaceutical compounds. Considering that heterocyclic systems have been the focus of intense synthetic efforts during the last decades, indenopyrazoles have thus been extensively screened. In this study, the authors have notably synthetized 16 compounds possessing a 4-substituted thiazole moiety, all prepared with indane-1,3-dione as the starting material, and two different series were developed (see Figure 157) in order to compare the impact of the fused indenopyrazole on the antimicrobial properties.

Several micro-organisms were tested such as Gram-positive bacteria (*S. aureus, B. subtilis),* Gram-negative-bacteria (*E. Coli)* and fungus *(Aspergillus niger, Candida albicans).* In this series of molecules, three molecules showed noticeable activities against *C. albicans, A. niger, S. aureus and E. Coli* (**528h**, **529d**, and **529h**). More precisely, antimicrobial activity of these molecules proved to be on par with that of the reference compound, i.e., Norfloxacin. Nevertheless, **528h** was less effective against *B. subtilis*, whereas the two others only showed moderate minimum inhibitory concentration (MIC) with this micro-organism. Noticeably, an improved antimicrobial activity was found for all compounds once cyclized as indenopyrazoles, and a higher activity was also determined for all molecules substituted with 4-chlorophenyl groups.

After highlighting the importance of this structure, the same team published a second work in which the synthetic strategy was modified compared to the previous work and seventeen 3-aryl-1-heteroarylindeno [1,2-*c*]pyrazol-4(1*H*)-ones **532** could be prepared with this new synthetic method [393]. Indane-1,3-dione **4** was used to perform Knoevenagel condensations with various aldehydes **368**, and the formation of the pyrazole cycle could be obtained by reaction of 2-hydrazinylbenzo[d]thiazole/2-hydrazinyl6-substituted benzo[*d*]thiazoles **530** and the Knoevenagel adduct **255** in stoichiometric amount, according to the reaction presented in Figure 158. Despite mild reaction conditions, the reaction yields remained low, ranging between 25 and 41%, even after optimization. Nonetheless, almost all of these new synthetized molecules showed an activity against four bacteria, Gram-positive and Gram-negative, but also against fungi.

In 2018, another multicomponent reaction involving indane-1,3-dione **4** was reported by Alsharif et al. for the design of an antimicrobial agent comprising nitrogen-based heterocyclic compounds [394]. The four-component reaction involving indane-1,3-dione **4**, 9-ethyl-9*H*-carbazole-3-carbaldehyde **533**, ethyl acetoacetate **534** and ammonium acetate **535** in the presence of a catalytic amount of piperidine could furnish ethyl 4-(9-ethyl-9*H*-carbazol-3-yl)-2-methyl-5-oxo-4,5-dihydro-1*H*-indeno [1,2-*b*]pyridine-3-carboxylate (ECPC, **536**), following a procedure previously reported in the literature (see Figure 159) [395]. This molecule was notably tested for its antibacterial activity, and for this study, *S. aureus Streptococcus pyogenes (S. pyogenes)* was selected as the Gram-positive bacteria and *E. Coli* as *Salmonella typhimurium (S. typhimurium)* as the Gram-negative bacteria. Biological activity of ECPC was also compared with that of tetracycine used as a standard for these tests. ECPC **536** showed an interesting antibacterial activity, since minimum inhibitory concentrations as low as 32 µg/mL were determined with the four bacteria, comparable to that required with the reference tetracycline.

In 2012, another series of heterocyclic molecules was synthetized using indane-1,3-dione **4** as the building block in an ionic liquid-catalyzed three-component condensation involving thiourea **537** and the appropriate aromatic aldehyde **368**, and the resulting molecules **538** were tested as antimicrobial agents (see Figure 160 and Table 38) [396]. In this work, the authors focused their attention on the pyrimidine scaffold, which is present in numerous natural physiologically active substances. The authors improved the biological activity of indane-1,3-dione derivatives by attaching a thione moiety to the pyrimidine group, this group being also known for its antifungal, antibiotic and antibacterial activities. Among the fourteen 4,6-diaryl- and 4,5-fused pyrimidine-2-thiones synthesized in this work, only three of them were prepared with indane-1,3-dione, the other compounds being variously substituted tetrahydrobenzo[*h*]quinazoline-2-thiones and pyrimidine-2-thiones. Among the three indane-1,3-dione derivatives, only **538c** showed a better antibacterial activity toward *B. subtilis, S. aureus, Pseudomonas aeruginosa (P. aeruginosa)* and *E. coli* and a better antifungal activity toward *A. niger, C. albicans, A. fumigatus* than the variously substituted tetrahydrobenzo[*h*]quinazoline-2-thiones and pyrimidine-2-thiones. In addition, among the eleven other 4,6-diaryl- and 4,5-fused pyrimidine-2-thiones, only two of them showed similar antimicrobial properties than **538c**, even if all of them have a less important zone of inhibition than the reference compound, namely Ampicillin trihydrate.

Finally, in 2016, another research group designed a series of spiro[indolo-3,10′-indeno [1,2-*b*]quinolin]-2,4,11′-triones **540** by means of a three-component condensation of enaminones **539**, isatin **231** and indane-1,3-dione **4** using a mixture of ethanol/water as the solvent and cerium ammonium nitrate (CAN) as the catalyst (see Figure 161) [397]. As the main advantage of this approach, spiro[indolo-3,10′-indeno [1,2-*b*]quinolin]-2,4,11′-triones **540** could be obtained with short reaction times, in high yields and by using a simple synthetic protocol. From this viewpoint, spiro[indolo-3,10′-indeno [1,2-*b*]quinolin]-2,4,11′-triones **540** reported in this work could be obtained by a greener approach than that previously reported in the literature [300,398,399,400,401]. Antimicrobial activities of these compounds have been tested on Gram-positive bacteria (*S. aureus* and *B. subtilis)* and Gram-negative bacteria (*E. coli* and *P. aeruginosa)* as well as their antifungal activity in two yeasts: *Candida albicans* and *Saccharomyces cerevisiae.* Among the twenty-two molecules synthetized in this work, all of them showed good antibacterial and antifungal activities, but none of them were effective against *P. aeruginosa*. Nevertheless, the best results were obtained for **540-IVc**. Notably, **540**-IVc exhibited the lowest MIC of 16 mg/mL against *S. aureus*, a MIC of 8 µg/mL against *B. subtilis* and a MIC of 64 µg/mL against *E. coli*. In the case of yeast, **540-IVc**, **540-IVk** and **540-IVn** showed the best inhibition ability against *C. albicans* and *S. cerevisiae* with inhibition zones of 15 and 16 mm, respectively. Furthermroe, **540-IVc** and **540-IVn** were also determined as being more efficient against *B. subtilis* and *S. aureus*, the inhibition zone reaching 22 and 24 mm, respectively. However, despite these promising results, none of the new compounds could surpass the inhibition diameters obtained with the reference compound, i.e., ciprofloxacin (26, 24 and 25 nm against *S. aureus*, *B. subtilis* and *E. Coli*, respectively).

### 5.2. Indane-1,3-Diones as Anticancer Agents

Cancer is a generic name that regroups all diseases where the proliferation of abnormal cells is observed. Since only half of the cancer patients do not recover with systemic chemotherapy or obtain only a partial recovery, the development of anticancer agents is thus the focus of intense research efforts [402]. This research was also supported by the current incapacity to cure cancers combined with longer life expectancy but also by the fact that cancer is the second leading cause of death.

The combination of *N*-heterocycles and indenones has been widely studied during the last 40 years for the design of various compounds exhibiting anti-cancer properties [403,404,405]. Inspired by Onychine, which is a natural biologically active azafluorenone, numerous synthetic azafluorenones have been prepared and identified as exhibiting promising cytotoxic, phosphodiesterase inhibitory, adenosine A2a receptor antagonistic, anti-inflammatory/antiallergic, coronary-dilating and calcium-modulating properties [406,407]. Following the course of these investigations, a series of indeno-heterocycles has been designed by an MCR involving indane-1,3-dione **4** as the building block, and a library of 33 indeno-heterocycles **541** could be obtained (see Figure 162). All these molecules were evaluated for their potential cytotoxic and apoptosis-inducing properties [407]. As specificities, all molecules reported in this work exhibited the same indenopyridine moiety in order to maintain the planarity of the structure. Planar structures are required in order for these molecules to act as DNA intercalators and topoisomerase inhibitors. Even if a similar synthetic protocol was used for the synthesis of the 33 molecules, different structures were nevertheless obtained. These compounds were notably tested for their cytotoxicity on Jurkat cells, a model for human T-cell leukemia, with Annexin-V/propidium iodide assay [408]. However, cytotoxicity of molecules **541** remained low since viability of the Jurkat cells was around 85–98% relative to the control experiments, and apoptosis induction was even lower, ranging between 1–6%. Aniline, pyrazole and triazole-based indeno-heterocycles **541** proved to be totally inactive for apoptosis induction. Only the pyrimidinedione-containing indeno-heterocycle showed a somewhat enhanced cytotoxicity, the cell viability being reduced to 80%. All attempts to mix the pyrimidinedione-containing indeno-heterocycle with the other compounds did not improve the cytotoxicity, evidencing that the pyrimidinedione moiety was the key-element to obtain an acceptable cytotoxicity. To obtain a deeper insight into the biological activity of this pyrimidinedione-containing indeno-heterocycle, this compound was compared to etoposide, which is an anti-cancer clinical drug agent known for its remarkable cytotoxic effect through topoisomerase II-dependent DNA cleavage. The two molecules could manifest good apoptosis-induction activities against Jurkat cells. Furthermore, the indeno-pyrimidine compound could demonstrate better cell-killing and apoptosis-induction activities than etoposide at low concentration (cytotoxic IC_50_ = 3 μM). Nevertheless, clinical tests still need to be performed with this pyrimidinedione-containing indeno-heterocycle, and the authors also highlight the important drawback of poor water solubility of this compound, which may affect its future use as anti-cancer agent.

Three years later, in the continuation of the pioneering work initiated by Manpadi et al., another series of 35 pentacycle-based indeno-heterocycles **544** and **545** was prepared in order to investigate the cell-killing and the apoptosis mechanism toward a panel of human cancer cell lines [409]. All these structures were inspired by camptothecin, which is extensively used in traditional Chinese medicine. As the main motivation of this study, the authors suggested the drug action of these molecules to operate via an intracellular oxidation of the dihydropyridine cycle of **544**, leading to the formation of the pyridine moiety **545** in situ (see Figure 163). To demonstrate this, all compounds were synthetized with a hydrogenated (“h”) and an aromatized (“a”) version of the indenopyridine core using the same three-component synthesis of that used by Manpadi. When no spontaneous oxidation was observed, an additional oxidation step with chloranil in DMF was required to obtain the aromatized structure. These molecules were further tested for their antiproliferative properties on a panel of human cancer cell lines (HeLa, Jurkat, MCF-7, A-549, Lovo, U373, SKMEL, PC3, MG-MID), representing many species of cancer such as T-cell leukemia, cervical, breast, or lung cancers. The authors demonstrated that the modification of the R group on the aromatic ring did not significantly modify the biological activity of the molecules, except that the molecule bearing R=H showed no cytotoxic activity. The biological tests also revealed that the dihydropyridines **544** do not have cytotoxic activities contrarily to the pyridine analogues **545**, supporting an intracellular oxidation process of the dihydropyridines **544** to the corresponding pyridines **545**.

Sulfonamides are well known to act as inhibitors of numerous human α-carbonic anhydrases [410]. Carbonic anhydrases are essential in humans, as these anhydrases are capable of catalyzing the reversible hydration of CO_2_ and allow for the respiration and the transportation of carbon dioxide within the human body. In 2012, the idea of Ghorab et al. was to combine the remarkable properties of sulfonamides to that of indenopyridines, which also possess anti-cancer properties [411]. To obtain these structures (**549**), a two-step synthesis involving indane-1,3-dione **4**, 4-aminobenzenesulfonamide **546** and an aromatic aldehyde **368** as the starting materials was performed, the two steps being both realized in EtOH as the solvent. Anti-cancer activity of the resulting 18 compounds **549a-549q** was tested in vivo against breast cancer cell-line (MCF 7) and their biological activities compared to that of doxorubicin, a reference drug. Among all compounds, **549d** exhibited an IC_50_ of 4.34 μM, lower than that of the reference doxorubicin (5.40 μM), evidencing the pertinence of the approach. Furthermore, **549d**, which is specifically substituted with a hydroxyphenyl group at the 4-position and a cyano group at the 3-position exhibited an increased cytotoxic activity compared to the other compounds, and this specific substitution is certainly at the origin of the higher potency of this molecule (see Figure 164).

Related to indenopyridines, indenopyrazoles are another family of structures showing important inhibiting tyrosine kinase activities, but also anti-cancer or anti-proliferative properties [412]. Nevertheless, for this purpose, indane-1,3-dione **4** was not directly used for their synthesis but a derivative containing a close scaffold: 2-(4-methoxybenzoyl)indan-1,3-dione **552**. Its synthesis is similar to that of compounds **4** and **67** described in Section 1 except that no heating was required for the last step to obtain the triketone molecule and, in a second step, the intramolecular cyclization of **552** with hydrazine hydrate could furnish the targeted indenopyrazoles **553**, as shown in Figure 165 [413]. In 2002, another library of indenopyrazoles mainly substituted at the 5-position of aniline and at the 3-position with various groups was examined as Cyclin-dependent kinase (CDK) inhibitors [414]. Cclin-dependent kinases play a key role in the cell cycle regulatory machinery and the replication process. Considering that the different indenopyrazoles can regulate the development of proliferative tumors such as cancers, these molecules have notably been tested against various cancer cell lines (HCT116, NCI-H460, PC-3, MiaPaCa-2, HT-29, HT-1080 and B16-F0). After selecting **553-k** for its good activity and selectivity against kinase targets (*CDK2/E*: IC_50_ = 13 nM, *CDK1/B:* IC_50_ = 44 nM), this molecule exhibited good cytotoxicity, particularly for HCT116 cells, which died not only by CDK inhibition but also by activation of the apoptotic machinery and by inhibition of Rb phosphorylation. This strong activity combined with a remarkable selectivity of cells offers very promising therapeutic applications, explaining why several patents have been established for structures close to that of **553-k** [412,415,416].

### 5.3. Indane-1,3-Dione as Building Block for Bioimaging Agents

Fluorescent imaging techniques play an important role in medicinal chemistry by enabling to localize tumors at the cellular level and in this aim; extensive works are performed to develop fluorescent dyes capable of target specific cells or ions [417]. With the aim at gaining a better understanding of all these complex diseases, multiple recent techniques have been developed to visualize tumors, and numerous fluorescent agents have been proposed with indane-1,3-dione, which remains a cheap and versatile building-block in chemistry.

One of the most popular imaging techniques is named aggregation-induced emission (AIE). As specificity, dyes exhibiting AIE properties are often weakly emissive in solution but highly emissive in the solid state due to a restriction of the intramolecular rotations that constitute non-radiative deactivation processes. In addition to enhancing the fluorescence intensity of dyes, AIE is also extensively used for bioimaging due to its biocompatibility, its high-fidelity imaging, high brightness and long-living excited state favorable for an efficient in-situ imaging [418]. In 2020, an interesting study demonstrated that beyond imaging, AIE dyes could also be used for the generation of reactive oxygen species (ROS), enabling to combine bioimaging and photodynamic therapy (PDT) [419]. However, the key point of this strategy relies in a perfect targeting of the infected lysosomes by AIE dyes. To achieve this goal, three molecules based on triphenylamines were designed, bearing morpholines as pendant groups. In these structures, triphenylamine was used as the electron donor for the design of the different push–pull dyes and indane-1,3-dione, malononitrile or the formyl group were used as the electron acceptors (see **MPAA 554**, **MPAN 555** and **MPAT 556** in Figure 166).

Among the three dyes, **MPAT 556** exhibited the most red-shifted absorption and emission of the series so that this dye was the focus of extensive works. The three dyes showed a remarkable fluorescence stability since upon irradiation with a laser for 20 min., a decrease of less than 10% of the initial fluorescence intensity was determined, far from the loss of 50% determined with the benchmark LysoTracker Green DND-26 [420]. Examination of the fluorescence properties of **MPAA 554**, **MPAN 555** and **MPAT 556** in water revealed these dyes to exhibit high stoke shifts ranging between 187, 160 and 188 nm for **MPAA 554**, **MPAN 555** and **MPAT 556**, respectively. For **MPAT 556** displaying the most red-shifted emission, fluorescence maxima at 673 nm in water and 675 nm in the solid state could be determined. Conversely, for the other dyes, emissions located at 530 and 614 nm could be respectively determined in water for **MPAA 554** and **MPAN 555.** For these reasons, the indane-1,3-dione adduct **MPAT 556** was selected for the biological tests in A549, HeLa and HepG2 cells. No significant decrease for the cells’ viability was detected after 24 h of incubation, even at high concentration (100 μM), demonstrating that **MPAT 556** can be used for long-term monitoring of cells without inducing cells apoptosis. In vivo experiments in zebrafish also furnished nice lysosome images with a good dot distribution during imaging of vertebrates. Finally, the ability of **MPAT 556** to promote ROS production was evidenced while using H2DCF-DA as the probe. In PBS buffer solutions, a 36-fold enhancement of the photoluminescence of H2DCF-CA was evidenced, demonstrating the production of ROS by **MPAT 556** upon green light irradiation. In HeLa cells, cell viability showed a remarkable decline, the cell viability reducing to 19% at 40 μM MPAT and after 10 min. of irradiation with a green light (see Figure 4).

In 2017, Gao et al. used AIE dyes for a completely different application, namely the lipid droplet-specific (LD) imaging and the dynamic movement tracking [421]. Lipid droplets (LDs) are involved in many metabolic processes. However, conventional dyes used to stain LDs suffer from numerous drawbacks such that new fluorophores are actively researched. Several families of molecules have been tested over the years, but the different fluorophores exhibited numerous limitations such as aggregation-caused quenching (ACQ), low stoke shift (40 nm), high background noise or even difficulty of preparation [422,423,424,425,426,427,428,429,430]. Consequently, the molecule selected in this study was a classical push–pull dye made of triphenylamine, as the electron-donating group was connected to indane-1,3-dione acting as the electron-withdrawing group and prepared by means of a Knoevenagel reaction. In addition of its AIE properties, **557** also exhibits what is named a twisted intramolecular charge transfer (TICT), resulting from restricted intramolecular rotations in the solid state. Based on both AIE and TICT properties, **557** was thus identified as a promising candidate for LD-specific imaging and movement tracking due to its high two-photon absorption cross-sections in the near infrared (NIR) range (see Figure 167). For this reason, **557** was notably used as fluorescent dye for near infrared (NIR) two-photon excited fluorescence (TPEF).

To evidence the AIE effect in **557**, UV–visible absorption and emission spectra were recorded in a THF/water mixture. A raise in the water fraction in THF from 0 to 70% led to a decrease in the fluorescence but also to a significant redshift not only of the emission spectrum (from 594 to 609 nm) but also of the absorption spectrum (from 478 to 489 nm), which can be attributed to TICT effects. A further increase in the water fraction in THF up to 99% induced a further redshift of the emission maximum up to 612 nm, resulting from the formation of nanoaggregates of average diameter of 119.6 nm, measured by light scattering analyses. Finally, **557** was tested on HCC827 and A549 cells. As shown in Figure 5, **557** could efficiently stain the cells with high signal-to-noise ratios. From the different tests, several advantages were determined for **557** such as:

A fast permeability;

An excellent selectivity for LD;

A high photostability;

A low cytotoxicity.

Finally, photostability of **557** was examined upon irradiation at 514 nm with a laser (7% power) for 10 min. As shown in Figure 6, the fluorescence intensity loss was less than 20%, evidencing the good photostability of this dye.

As we saw previously, the selectivity of dyes to target the right cells is a primordial need, especially when photodynamic therapy (PDT) is envisioned in complement to cell visualization. In 2019, Mondal et al. performed an MCR using indane-1,3-dione **4** as a component to elaborate a series of spiro-heterocyclic compounds **561–564** (see Figure 168) that was subsequently tested as chromophores for the selective detection of Zn^2+^ ion and cell imaging [431]. Regarding specificity, this synthesis was realized by means of a microwave-assisted multi-component reaction (MWAMCRs); the reaction was carried out in EtOH, and the products could be obtained in pure form without chromatography, making this approach highly biocompatible.

Nine of the compounds were selected for further tests aiming at evaluating the possibility to coordinate Zn^2+^ ion, determining the selectivity toward other ions other than Zn^2+^ or the fluorescence emission at physiological pH. Among the nine dyes, one of them (i.e., **564f**) was even selected for further in vivo tests on human hepatocellular liver carcinoma cells (HepG2 cells). This chemosensor could be successfully used as an intracellular Zn^2+^ imaging agent due to its remarkable cell permeability properties, where this molecule exhibited a fluorescence comportment only when coordinated with Zn^2+^ even in vivo. Cytotoxicity tests also revealed **564f** to be weakly toxic since the cell viability was higher than 90% at 10 µM. Thus, **564f** can be effectively used in vivo as chemosensors for the detection of Zn^2+^ cations. A mechanism supporting the enhancement of fluorescence upon coordination to Zn^2+^ cations was also proposed by the authors and is depicted in Figure 169.

### 5.4. Indane-1,3-Dione in Neurology Drugs

The brain is the most complex organ in humans, and its working principle is still not fully understood yet, which explains the lack of medicines specific to this organ, despite all the research performed in this field. Moreover, the eldering of the populating on Earth will cause a significant increase in the neurodegenerative diseases that usually occur after 65 years, so that the demand on treatments for these illnesses will increase in the coming decades [432,433]. Parkinson’s disease (PD) is one of the most important neurodegenerative diseases and causes an important loss of dopamine in the brain, altering motor function cognition and the mood of the patient. Common treatments to PD are antagonists of two brain receptors, namely A_2a_ and A_1_. However, it is still unknown yet if either one of the two brains receptors plays a more important role than the other in the disease [434].

In this context, Shook’s team published a series of arylindenopyrimidines **565** that were developed as potential dual A_2A_/A_1_ antagonists and that could be potentially used for the treatment of AD [435]. These arylindenopyrimidines **565** were synthetized through a two-step synthesis using indane-1,3-dione **4** as the starting material (see chemical structures in Figure 170). In vitro and in vivo tests performed with eight molecules revealed unequal results depending on the substitution pattern. Notably, the different tests revealed the necessity to let the NH_2_ group be unsubstituted, whereas the substitution of the pendant aromatic group had almost no incidence on the in vivo and in vitro activity. Furthermore, molecules substituted with morpholine or pyrrolidine groups showed the highest in vivo activity in mouse catalepsy.

To improve the biological activity, two different syntheses were developed to obtain substitutions on the heterocycle next to the indane group. The 12-step synthesis developed by the authors could give access to twelve compounds **577** substituted at the 9-position, and the seven-step synthesis developed in parallel could furnish 12 other compounds **584** substituted at the 8-position (see Figure 171).

Comparison between the two series of analogs **577** and **584** revealed an improved biological activity for all molecules substituted at the 8-positions compared to those bearing a substitution at the 9-positions. Notably, a difference in the mouse catalepsy as high as 0.2 and 10.7 mg/kg for the ED_50_ for the regioisomers **577-(1)** and **584-(1)** bearing a piperidine substituent could be determined. Furthermore, several adducts, particularly those containing cyclic amines and heterocyclic functions attached with an alkyl secondary amine showed ED_50_ in the micro-scale for the mouse catalepsy. These adducts also showed decent in vitro activity so that these structures can be cited as promising candidates for AD treatments (see Figure 172). Nevertheless, the lack of the commercially available substituted indane-1,3-diones allowing access to these structures is relatively complex and costly.

Furthermore, these structures remain promising, as exemplified with JNJ-40255293 (**585**) for which preclinical tests have been carried out (see Figure 173) [436]. These additional tests revealed a better selectivity of the JNJ-40255293 antagonist toward the A2a receptors than the A1 ones. Unfortunately, preclinical tests were rapidly abandoned due to genotoxicity, neuronal necrosis and edema observed in several animals [437].

In 2002, more promising results were obtained with another series of indeno-fused structures, namely arylindenopyridines **588** and **593** that were patented for AD treatments (see Figure 174) [438]. To support the biological efficiency, the authors suggested a binding of arylindenopyridines onto the A2a receptors. Anti-inflammatory activity was also identified, with phosphodiesterase (PDE)-inhibiting activity. The starting materials for these syntheses were various 1,3-indanedione adducts (**586** or **590**), such as 5-nitroindane-1,3-dione (see Figure 138). The MCR used to access to these structures involved in the first step a Knoevenagel reaction between the indane-1,3-dione derivative **586** or **590** with the appropriate aldehyde **368**, followed by a condensation of guanidine carbonate **588** to the push–pull compound **587** to form the indenopyrimidine **589**. In the case of **491**, an aromatization with DDQ was required, providing **593**. The selectivity toward the A_2a_ receptor and A_1_ were tested in vitro, and it turned out that some of these molecules could exhibit K_i_ lower than 50 nM, particularly with the A_2a_ binding. Despite these interesting preliminary results, no in vivo tests were performed with these molecules.

In 2011, a series of tricyclic 3,4-dihydropyrimidine-2-thiones **595** was proposed as a potential A_1_ receptor and TRPA1 antagonists. The TRPA1 channel is implicated in numerous inflammatory or neuropathic pains [439]. These molecules were notably prepared via a Biginelli reaction with indane-1,3-dione **4** due to which the effects of the substitution of the phenyl groups along with the replacement of urea by thiourea in the cyclization reaction were studied. According to the in vitro tests performed on human and rats cells, experimental results revealed that all compounds prepared with thiourea were more biologically active than those prepared with urea. Influence of the stereochemistry is also clearly evidenced, as shown for compounds **595-(3)** and **595-(4)**. Thus, for **595-(2)**, a half maximal inhibitory concentration hTRPA1 IC_50_ of 0.075µM was determined for the *S,R*-stereoisomer, whereas for the *S,S*-stereoisomer **595-(3)**, an hTRPA1 IC_50_ higher than 10 µM was determined (see Figure 139). The *meta*-substitution on the phenyl ring could induce a significant increase in the activity (see Figure 175).

Nevertheless, only in vitro tests were performed with these molecules, and the authors highlighted some drawbacks such as a low solubility, a poor metabolic stability and the potential toxicity of thiourea used to prepare these molecules [440].

In 2015, Ahmed et al. described a solvent-free three-component synthesis based on 1,3-indanedione as the starting material and developed a series of 14 potential anticonvulsants compounds with this strategy [243]. Seizure is a transient occurrence of signs and/or symptoms due to an abnormal excessive or synchronous neuronal activity in the brain [441]. This condition is often affiliated with epilepsy, which concerns about 0.5–1% of the population. Moreover, serious improvement needs to be performed concerning their efficiency but also to address the important issue of side effects [442]. The authors principally acted on one factor, namely the substitution on the pendant phenyl group. After synthesis, each molecule was tested via an anticonvulsant evaluation with the maximal electroshock (MES) method and compared with the results obtained with a standard drug, i.e., phenytoin, along with their toxicity with the Rotorod test. Most of the compounds designed in this study seemed to have positive effects on seizures ranging from moderate to good activity, particularly **222d**, **222e**, and **222j** (see Figure 176), which proved to be significantly active at a dose of 40 mg/kg. Moreover, the Rotarod tests showed no toxicity for all the compounds tested. However, none of them were as efficient as the reference compound phenytoin.

Among the neurodegenerative diseases, Alzheimer’s disease (AD) is the most common one, but this disease is also badly treated. During the last 25 years, only five medicines were approved for AD treatment with two different strategies of action. Thus, cholinesterase inhibitors (ChEIs) such as tacrine (1), donepezil (2), rivastigmine (3) and galantamine (4) were proposed in parallel to N-methyl-D-aspartate (NMDA) antagonists such as memantine (5) [443]. Nonetheless, none of them can repair the damage or delay the disease progression, which could constitute a major improvement in this field. The aggregation of a small peptide named amyloid β (Aβ) is associated with AD, and the inhibition of both cholinesterase sites (acetylcholinesterase, AChE, and butyrylcholinesterase, BChE) may prevent this aggregation [444].

In 2018, Tanoli et al. developed a series of tricyclic fused ring systems exhibiting activity against both acetylcholinesterase (AChE) and butyrylcholinesterase (BChE). Precisely, this study was supported by previous works reported in the literature, demonstrating that other tricyclic structures based on pyrimidine or quinazoline could act as AChE and BChE inhibitors [445]. In order to access the pyrimidine-fused rings, indane-1,3-dione **4** was used in a Biginelli reaction along with thiourea **597** and benzaldehyde **368** (Figure 177).

The dihydropyrimidine (DHPM) scaffold is a potential inhibitor of cholinesterases so that in vitro tests were carried out. Interesting results were obtained with these molecules since the introduction of the indanone-pyrimidine fused ring could lead to an improved inhibition activity compared to tacrine and donepezil for both electric eel AChE (IC_50_ = 0.09 µM) and equine serum BChE (IC_50_ = 1.04 µM) assays. These results are close to that obtained with the commercially available Donepezil with IC_50_ *ee*AChE = 0.05 µM and IC_50_ *eq*BChE = 5.4 µM. Docking studies on this molecule have also exposed the well accommodation of this molecule into the bottom of the gorge and the importance of the hydrogen bonding with its environment, such as Ser200, or even the phenyl ring at the 4-position, which forms π-π stacking interactions with trp84 (see Figure 7).

In 2010, the good activity of 2-[(2-(4-chlorophenyl)hydrazinyl) methylene]-1*H*-indene-1,3(2*H*)-dione **605f** as a novel inhibitor capable of prevent amyloid aggregation (IC_50_ = 23µM) [446] motivated the group of Campana to obtain a deeper insight into the biological activity of this molecule and in this aim a series of indane-2-arylhydrazinylmethylene-1,3-diones **605a–f** and indol-2-aryldiazenylmethylene-3-ones **608a–m** (see Figure 178) [447].

In this work, the authors clearly highlight a significant increase in the biological activity with the molecules prepared with indane-1,3-dione compared to those prepared with 1,3-cyclopentanedione. Nonetheless, even if all the compounds exhibited decent to good AB aggregation processes, none of them could overcome the reference quercetin (IC_50_ = 0.8 µM).

### 5.5. Indane-1,3-Dione as Anticoagulant Drugs

One of the oldest utilizations of indanedione derivatives in medicine concerns the anticoagulant properties. Since the 1940s, several authors reported the insecticidal properties of acylated indane-1,3-diones (see Figure 179), especially for houseflies [154]. Later on, this toxicity has been attributed to an anticoagulant property, especially marked for acylated indane-1,3-diones (see Figure 179) [448]. However, this activity can also be used as treatment for vitamin K antagonists (VKAs), as observed for phenindione derivatives (see Figure 179) [449]. However, pheindione is not so much used anymore and was replaced by its homologue, fluindione, or other coumarins VKA due to side effects such as hypersensitivity identified for pheindione [450]. However, despite a proven efficiency as vitamin K epoxide reductase, fluindione **609b** is used almost only in France because of a lack of study data on old people [451].

The synthesis of these indane-1,3-dione derivatives is actually not performed anymore starting from the unsubstituted indane-1,3-dione, but a recent study uses a similar structure with indane-1,3-dione as the starting material [452]. Nonetheless, this molecule is only an intermediary of reaction for the synthesis of 2,3-disubstituted indoles.

Concerning the synthesis of acylindanediones **613**, the standard synthetic strategy is quite similar to that used for indane-1,3-dione **4** since it consists in a Claisen condensation using an alcoolate as the base (see Figure 180) [453]. However, an alternative was proposed in 2018 by Larsen et al. involving indane-1,3-dione **4** as the starting material with the possibility to afford 25 different C2 acylated 1,3-indandiones **616** (see Figure 181) [454]. This strategy, which is quite efficient, makes use of an easy set up and producing the molecules in high yields using 1-ethyl-(3-(3-dimethylamino)propyl)-carbodiimide (EDCI) and 4-dimethylaminopyridine (DMAP) as the coupling agents. Furthermore, all tests made with this new strategy have only been performed at a small scale compared to the initial synthesis.

These type of indanes are more commonly used as rodenticides than as for pharmaceutical purposes. All commercially available rodenticides are either 4-hydroxycoumarins or indane-1,3-diones and even if coumarins are more widely used, a resistance has recently been evidenced toward derivatives such as Warfarin. At present, less resistances have been identified toward indane derivatives. Chlorophacinone, which belongs to the first generation of rodenticides, is still commonly used in many countries. Even if the second generation is more effective against rodents, their impact on the environment is quite significant with multiple poisoning detected in cats, dogs or even otters [455,456].

## 6. Conclusions-Perspectives

Indane-1,3-dione is a versatile molecule that has found applications in numerous research fields. Directly related to the broadness of applications, numerous synthetic routes have been examined first to prepare indane-1,3-dione, the corresponding derivatives bearing various substituents. If indane-1,3-dione is an excellent electron acceptor for the design of various push–pull dyes, the most popular use of indane-1,3-dione is undoubtedly in multicomponent or domino reactions to prepare polycyclic structures for biological activities. Considering the remarkable biological activity of indane-1,3-dione derivatives, future work will certainly be more focused on these biological applications than on the optoelectronics applications. Numerous electron acceptors have already been prepared with this scaffold, limiting the possibility of new developments.

## Data Availability

No data available.

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
