# Peer review of "Indane-1,3-Dione: From Synthetic Strategies to Applications"

_molecules, 2022, doi:10.3390/molecules27185976_

Round 1
Reviewer 1 Report
This review summarizes numerous studies on synthesis and application of indane-1,3-dione and its derivatives. The authors also describe reactivity of indane-1,3-dione in detail to be understandable for readers. The topic has attracted much attention in the field of organic and material chemistry and is appropriate for Molecules as Review. Therefore, this manuscript can be accepted for publication after minor revision. The authors should consider the following comments.
Points to be attended to are described below:
(1) Page 10, Scheme 10: structure of compound 70 is wrong. Please correct it.
(2) Page 22, Scheme 25: structure of compound 158 a is wrong (two C=C bonds should be single bond). Please correct it.
(3) Page 90, Scheme 95: mechanism of the domino reaction does not correspond to the scheme in Table 26 on Page 89, which makes it complicated to understand. Please confirm this point.
(4) Page 93, the second line from the bottom of the paragraph: … the aromatic aldehyde 428 should be … the aromatic aldehyde 429. Please correct it.
(5) Page 102, Scheme 107 and Scheme 108: explanation about Scheme 107 and Scheme 108 are not found in the text on Page 101.
(6) Page 108, line 4: compound No. of the Seyferth-Gilbert reagent is 468, not 457. Please correct it.
(7) Page 110, line 6: the authors described that a selection structures (460, 461-465, 466, 467, 468-487) is presented in Scheme 114, however, only compounds 470-487 are shown. Why?
(8) Page 110, the fourth line from the bottom of the paragraph: a new series of 21 dyes 468-487 seems to be wrong (20? dyes). Please confirm it.
Author Response
This review summarizes numerous studies on synthesis and application of indane-1,3-dione and its derivatives. The authors also describe reactivity of indane-1,3-dione in detail to be understandable for readers. The topic has attracted much attention in the field of organic and material chemistry and is appropriate for Molecules as Review. Therefore, this manuscript can be accepted for publication after minor revision. The authors should consider the following comments.
Points to be attended to are described below:
(1) Page 10, Scheme 10: structure of compound 70 is wrong. Please correct it.
We do agree with this comment. The number under the chemical structure was wrong. It was replaced by 32a. Additionally, the final molecule was not 31 but 32. The scheme has been corrected.
(2) Page 22, Scheme 25: structure of compound 158a is wrong (two C=C bonds should be single bond). Please correct it.
Chemical structure of 158a has been corrected.
(3) Page 90, Scheme 95: mechanism of the domino reaction does not correspond to the scheme in Table 26 on Page 89, which makes it complicated to understand. Please confirm this point.
In fact, Scheme 95 corresponds to the reaction presented in the Scheme of Table 89. However, in Scheme 95, the indanedione moiety has been removed for clarity and replaced by E, E’. The reaction occurring on the rest of the molecule can be better visualized.
(4) Page 93, the second line from the bottom of the paragraph: … the aromatic aldehyde 428 should be … the aromatic aldehyde 429. Please correct it.
Numbering has been corrected.
(5) Page 102, Scheme 107 and Scheme 108: explanation about Scheme 107 and Scheme 108 are not found in the text on Page 101.
Explanations of the two schemes have been added.
(6) Page 108, line 4: compound No. of the Seyferth-Gilbert reagent is 468, not 457. Please correct it.
Numbering has been corrected.
(7) Page 110, line 6: the authors described that a selection of structures (460, 461-465, 466, 467, 468-487) is presented in Scheme 114, however, only compounds 470-497 are shown. Why?
This is again a problem of numbering. The correct numbers are 470-497 and not 460-487. In fact, in this review, more than 500 structures are presented. During writing, the numbers have been modified several times.
(8) Page 110, the fourth line from the bottom of the paragraph: a new series of 21 dyes 468-487 seems to be wrong (20? dyes). Please confirm it.
Numbering of molecules has been corrected.
Reviewer 2 Report
The review by Prof. Frédéric Dumur and coworkers provides the Indane-1,3-dione synthetic strategies and applications. The presentation of the indane-1,3-dione synthetic and applications is clean and well-written, and the references are sufficient. I would recommend this manuscript for publication at molecules after minor revision:
1. Most of the figures & schemes are crowded. Authors might consider expanding it from left to right. Most of the schemes should reorganized.
2. Scheme 4, Can you combine the compounds 36a, 36b, 36c, 36d, 38, 40,42, 44, 46 ,48 as 4,5,6,7-subtituted Indane-1,3-dione, so the scheme is clear and brief.
3. Page 8, Scheme 6. Compound 51 should be ‘hydrazine carbothioamide’, then the compound 52 should be rewrite.
4. Page 22, Scheme 25, the compound 158a under the H2, Pd/C condition, it should be alkane, not contain double bond.
5. Page 55, Scheme 57, please make the scheme much clearer.
6. Make all the scheme consistent.
7. Scheme 57. Not clear, please draw a better schemes, many mechanism schemes are picture, please use Chemdraw style,
8. Please added the compounds number from scheme 143 to 145.
Author Response
The review by Prof. Frédéric Dumur and coworkers provides the Indane-1,3-dione synthetic strategies and applications. The presentation of the indane-1,3-dione synthetic and applications is clean and well-written, and the references are sufficient. I would recommend this manuscript for publication at molecules after minor revision:
- Most of the figures & schemes are crowded. Authors might consider expanding it from left to right. Most of the schemes should reorganized.
In fact, sizes and positions of the figures are imposed by the template of molecules. Unfortunately, we are not allowed to modify the template.
- Scheme 4, Can you combine the compounds 36a, 36b, 36c, 36d, 38, 40,42, 44, 46 ,48 as 4,5,6,7-subtituted Indane-1,3-dione, so the scheme is clear and brief.
We don’t agree with this comment. In fact, due to the variety of substituents, from our viewpoint, it’s clearer to present the different molecules separately than combined in a generic structure. Therefore, presentation of scheme 4 is maintained.
- Page 8, Scheme 6. Compound 51 should be ‘hydrazine carbothioamide’, then the compound 52 should be rewrite.
We thank the reviewer for its comments. Chemical structures of 51 and 52 have been corrected in the Scheme.
- Page 22, Scheme 25, the compound 158a under the H2, Pd/C condition, it should be alkane, not contain double bond.
Chemical structure of 158a has been corrected.
- Page 55, Scheme 57, please make the scheme much clearer.
This scheme is extracted from reference 260. Quality of the figure has been improved as suggested.
- Make all the scheme consistent.
The different schemes are of different sizes. Unfortunately, we have no liberty to modify the size of the schemes with the template. All drawing have been prepared by the same authors. Therefore, the police and size of molecules are consistent between them.
- Scheme 57. Not clear, please draw a better scheme, many mechanism schemes are picture, please use Chemdraw style.
Scheme 57 has been modified as suggested. We do agree that numerous mechanisms are figures. These figures have been extracted from the corresponding article. Their origins are provided under each figure.
- Please added the compounds number from scheme 143 to 145.
Numbers have been added to the different molecules.